# Targeting IFNα to tumor by anti-PD-L1 creates feedforward antitumor responses to overcome checkpoint blockade resistance

Yong Liang[1,2], Haidong Tang [2,3], Jingya Guo[1,4], Xiangyan Qiu[3], Zecheng Yang[4], Zhenhua Ren[3], Zhichen Sun[1], Yingjie Bian [1], Lily Xu[5], Hairong Xu[1], Jiao Shen[4], Yanfei Han[1], Haidong Dong [6], Hua Peng[1] & Yang-Xin Fu [3]

Many patients remain unresponsive to intensive PD-1/PD-L1 blockade therapy despite the presence of tumor-infiltrating lymphocytes. We propose that impaired innate sensing might limit the complete activation of tumor-specific T cells after PD-1/PD-L1 blockade. Local delivery of type I interferons (IFNs) restores antigen presentation, but upregulates PD-L1, dampening subsequent T-cell activation. Therefore, we armed anti-PD-L1 antibody with IFNα (IFNα-anti-PD-L1) to create feedforward responses. Here, we find that a synergistic effect is achieved to overcome both type I IFN and checkpoint blockade therapy resistance with the least side effects in advanced tumors. Intriguingly, PD-L1 expressed in either tumor cells or tumor-associated host cells is sufficient for fusion protein targeting. IFNα-anti-PD-L1 activates IFNAR signaling in host cells, but not in tumor cells to initiate T-cell reactivation. Our data suggest that a next-generation PD-L1 antibody armed with IFNα improves tumor targeting and antigen presentation, while countering innate or T-cell-driven PD-L1 upregulation within tumor.

[1] Chinese Academy of Sciences Key Laboratory of Infection and Immunity, Institute of Biophysics, Chinese Academy of Sciences, Beijing 100101, China. [2] School of Pharmaceutical Sciences, Tsinghua University, Beijing 100084, China. [3] Department of Pathology, University of Texas Southwestern Medical Center, Dallas, TX 75235, USA. [4] University of Chinese Academy of Sciences, Beijing 100049, China. [5] Department of Biology, Wellesley College, Wellesley, MA 02481, USA. [6] Departments of Urology and Immunology, College of Medicine, Mayo Clinic, Rochester, MN 55905, USA. These authors contributed equally: Yong Liang, Haidong Tang, Jingya Guo.  Correspondence and requests for materials should be addressed to H.T. (email: hdtang@mail.tsinghua.edu.cn) or to H.P. (email: hpeng@moon.ibp.cas.cn) or to Y.-X.F. (email: Yang-Xin.Fu@UTSouthwestern.edu)

Programmed cell death protein 1 (PD-1) is a critical immune checkpoint during which inhibitory signaling is transmitted to T cells in order to prevent autoimmune responses. In tumors, the PD-1 ligand (PD-L1) is upregulated to evade immune responses[1]. PD-1/PD-L1 blockade therapy can induce an unprecedented, enduring response in patients with a variety of cancers[2,3]. However, objective responses are only observed in a small portion of patients. Tumor resistance to PD-1/PD-L1 blockade therapy after the initial response has additionally drawn increased concern, but the mechanisms are poorly defined[4]. It has become a top priority to understand why certain tumors are unresponsive to or develop resistance against PD-1/PD-L1 blockade therapy.

It has been suggested that effective tumor control by PD-1/PD-L1 blockade therapy is due to the release of immune-suppressive signaling in T cells. A sufficient number of tumor-infiltrating lymphocytes (TILs) has reportedly been correlated with a better response to PD-1/PD-L1 blockade therapy[5]. However, even in the presence of heavy lymphocyte infiltration, PD-1/PD-L1 blockade therapy alone might not be effective to (re-)activate tumor-specific T cells[6]. In these situations, blockade of other negative co-inhibitors or upregulation of stimulatory signals may be required to induce T-cell (re-)activation[7,8]. However, the identity of the signal(s) that effectively boost T-cell immunity is still under debate.

Several studies have shown that type I interferons (IFNs) play critical roles in the tumor control by promoting dendritic cell (DC) cross-priming to (re-)activate T cells[9–11]. However, the expression of type I IFN inside the tumor microenvironment (TME) is limited or suppressed. In fact, the increased degradation of DNA and antigen within the TME and the absence of cGAS-STING pathway signaling in some tumor cells might limit innate sensing and type I production[9,12]. Furthermore, IFNs are most potent cytokines to induce PD-L1, dampening the subsequent T-cell response against the tumor via a negative feedback effect[13–15]. To overcome these limitations, we arm anti-PD-L1 antibody with IFNα to simultaneously target both PD-L1 and IFN-receptor. More effective targeting of IFNα to tumor tissues is observed. IFNα-anti-PD-L1 treatment increases antigen cross-presentation and overcomes PD-L1-mediated immune suppression. Hence, we have developed a next-generation anti-PD-L1 antibody that may coordinate both PD-1-brake releasing and accelerating (re-)activation of T cells for tumor control.

## Results

### IFNα delivered to tumor overcomes PD-L1 blockade resistance.
A recent study has shown that clinical response in patients treated with checkpoint blockade correlates with the ratio of T-cell invigoration to tumor burden[6]. We consistently found that smaller A20 tumors ($< 50\ mm^3$) contained a higher ratio of $CD8^+$ T cells, compared to advanced tumors ($> 100\ mm^3$) (Supplementary Fig. 1a). Also, the anti-PD-L1 antibody showed effective tumor control in small A20 tumors (Fig. 1a), while the antitumor effects were dramatically reduced when the tumors became more established (Fig. 1b). Advanced tumors may have developed multi-mechanisms to inhibit antitumor immune responses[4]. When comparing the T-cell activation in small vs. large tumors, we observed a similar outcome; PD-L1 blockade induced robust tumor-specific T-cell increase in small tumors, while the same treatment had limited effects on those in advanced tumors (Fig. 1c). PD-L1 blockade also activated the effector function of T cells in small tumors, as increased $IFN\gamma^+TNF\alpha^+$ T cells were detected after treatment, but not in advanced tumors (Supplementary Fig. 1c–f). These data suggest that the T cells might be more exhausted in advanced tumor, and could not be

activated efficiently by checkpoint blockade alone. Type I IFN is a potent cytokine for increasing cross-presentation to cytotoxic T cells. Moreover, IFNA1 expression level positively correlates with better survival in human cancer patients (Supplementary Fig. 2a). Consequently, we explored whether providing additional type I IFN to activate T cells could improve PD-L1 blockade therapy. We produced IFNα-Fc; Fc fusion is reported to increase half-life in vivo and binding affinity through dimerization. A20 Tumors were treated with a combination therapy of PD-L1 blockade and IFNα-Fc. While tumors were tolerant to anti-PD-L1 treatment alone, they were not controlled efficiently by IFN-Fc treatment alone either (Fig. 1d). The combination therapy induced remarkable antitumor effects with a complete tumor eradication in all the treated mice (Fig. 1d). Similar synergistic effects were found in the MC38 tumor model (Fig. 1e), and notably the combination therapy but not single treatment alone induced 60% of complete tumor clearance (Supplementary Fig. 2b). Type I IFN signaling seems to function locally in the TME, as tumor-targeting of IFNα through intratumoral injection effectively controlled tumor growth (Fig. 1f). In contrast, the effects completely disappeared when IFNα was delivered via intravenous injection, through which it is not possible to reach sufficient serum concentrations to enable comparable intra-tumor accumulation achieved via i.t. injection. Taken together, our data suggest that type I IFN synergizes with PD-L1 blockade, leading to improved control for advanced tumors, but needs to be targeted locally to the TME in order to achieve optimized antitumor effects.

### Construct armed PD-L1 antibody to deliver IFNα into tumor.
Local delivery of type I IFN for tumor therapy is reported in mouse studies and limited clinical trials[16–18], but may not be feasible for most patients. Systemic delivery of type I IFNs is usually used clinically, but high-dose need to be administrated to achieve high-serum concentration for effective therapeutic effect accompanied with dose-limiting toxicities. Arming tumor-targeting antibodies with cytokines has been shown to be a potent strategy for the local delivery of immunomodulatory molecules[19–22]. However, it is difficult to identify tumor-specific molecules for therapeutic targeting. PD-L1 has been reported to be highly expressed in tumor tissues[1]. Our recent study shows that the anti-PD-L1 antibody accumulates specifically in PD-L1$^+$ tumors[23]. Moreover, in addition to their antitumor functions, IFNs are the most potent inducers of PD-L1 expression, thus dampening T-cell responses to tumors. To reverse this offsetting effect and achieve a mutual promotion of immune (re-)activation in the TME, we propose that arming PD-L1 antibodies with IFNα can further upregulate PD-L1 expression in tumor tissues and thus lead to increased antibody accumulation[1,13]. To test this hypothesis, we generated fusion proteins containing a single-chain variable fragment of anti-PD-L1 antibody [scFv(PD-L1)] and IFNα in either homodimer or heterodimer format (Fig. 2a). Sodium dodecyl sulfate polyacrylamide gel electrophoresis and capillary electrophoresis (CE) results showed that the purified proteins have a purity >95% (Supplementary Fig. 3a, b). To evaluate the binding of the resulting IFNα-anti-PD-L1 fusion protein, we tested their affinities for either PD-L1 or IFNAR. A20 cells are positive for both PD-L1 and IFNAR1. So we knocked out one receptor and tested binding to the other. In PD-L1-expressing IFNAR1$^{-/-}$ A20 cells, the fusion protein bound with a similar affinity to that of the anti-PD-L1 antibody (Fig. 2b). In IFNAR-expressing PD-L1$^{-/-}$ cells, the binding of the heterodimer was reduced in comparison to that of IFNα-Fc or to that of the homodimer (Fig. 2c). The binding affinities to PD-L1 determined by Biolayer interferometry (BLI) showed that heterodimer and

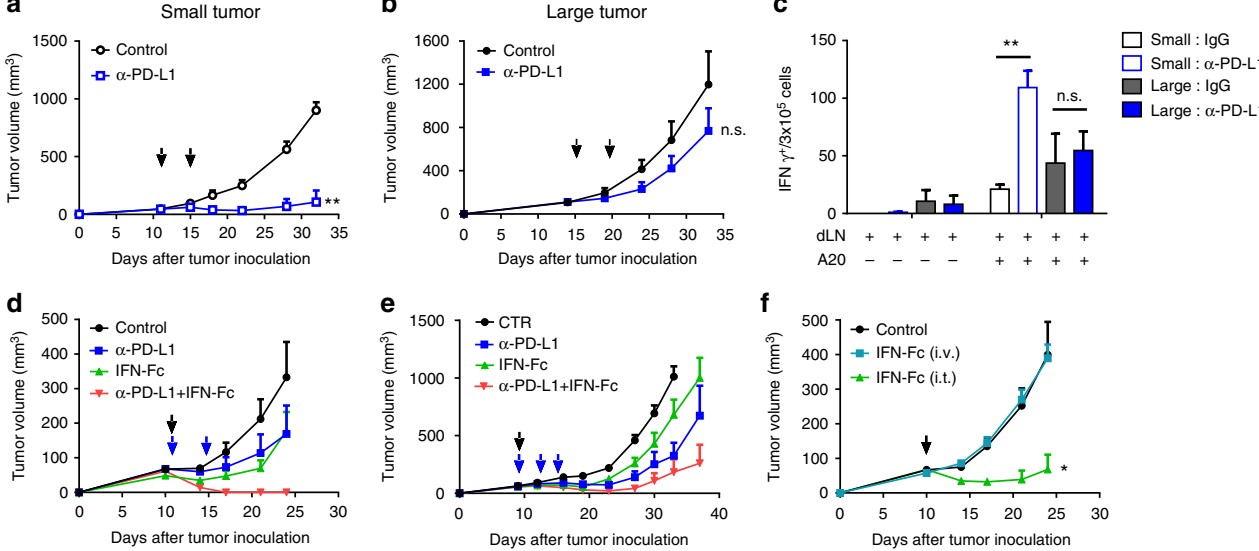

**Fig. 1** Local delivery of IFNα overcomes resistance to PD-L1 blockade in advanced tumors. **a** Balb/c mice ($n = 3$) were inoculated with $3 \times 10^6$ A20 cells. Mice bearing early-stage tumors ($< 50$ mm³) were treated intraperitoneally (i.p.) with 200 μg of anti-PD-L1 on days 11 and 15. **b** Mice ($n = 4$) bearing advanced A20 tumors ($> 100$ mm³) were treated with anti-PD-L1 on days 15 and 19. Tumor growth was measured twice per week. **c** Mice were treated as in **a** and **b**. Three days after treatment, cells from draining lymph nodes were isolated and co-cultured with or without irradiated A20 cells for 2 days. The IFNγ ELISPOT assay was performed. **d** A20 tumor-bearing mice ($n = 5$) were treated with 200 μg of anti-PD-L1 i.p. on days 11 and 15 and/or 25 μg IFNα-Fc intratumorally (i.t.) on day 11. **e** C57BL/6 mice ($n = 5$) were inoculated with $5 \times 10^5$ MC38 cells. Mice were treated with 200 μg of anti-PD-L1 i.p. on days 9, 12, and 15, and/or 25 μg of IFNα-Fc i.t. on day 9. The growth curves of MC38 tumors are shown. **f** A20 tumor-bearing mice ($n = 5$) were treated i.t. or intravenously (i.v.) with 25 μg of IFNα-Fc on day 11. Black and blue arrows indicate treatment with IFNα-Fc and anti-PD-L1, respectively. Data indicate mean ± SEM and are representative of at least two independent experiments. * $p < 0.05$; **$p < 0.01$; n.s. not significant

homodimer have $K_D$ values in the nM range (2 nM or 0.2 nM) (Supplementary Fig. 3c). The type I IFN bioactivity in the fusion proteins was identified through anti-viral infection biological assay. IFNα-Fc protein could inhibit VSV-GFP (vesicular stomatitis virus expressing green fluorescent protein) infecting L929 cells in a dose-dependent manner. While at the same efficient concentration, both homodimer and heterodimer protein showed higher inhibition efficiency compared to that of the control IFNα-Fc protein (Fig. 2d). Collectively, these data suggest that both the homodimer and heterodimer formats of IFNα-anti-PD-L1 fusion protein can bind to PD-L1 while maintaining robust IFNα bioactivity.

**Targeted delivery of IFNα by anti-PD-L1 controls large tumor.** We investigated whether the IFNα-anti-PD-L1 fusion protein can control tumor growth in vivo, given its potency. Mice bearing advanced A20 tumors were treated with fusion protein intratumorally (i.t.). While the anti-PD-L1 antibody failed to control tumor growth, both the heterodimer and homodimer forms of the IFNα-anti-PD-L1 fusion protein overcame anti-PD-L1 resistance and induced complete tumor regression in most of the treated mice (Fig. 2e). To test the targeting effects of the fusion protein, A20 tumor-bearing mice were treated with fusion protein systemically. To our surprise, although the homodimer showed higher binding affinity to IFN-receptor and more potent anti-viral activity in vitro (Fig. 2c, d), only the heterodimer was able to control tumor growth in vivo (Fig. 2f and Supplementary Fig. 4a, b). Similar effects were observed in the MC38 model (Fig. 2g). We wondered whether this difference was due to the drug's kinetics in vivo. This was indeed the case, the tumor tissues contained a much higher concentration of the drug in its heterodimer form than in its homodimer form (Fig. 2h); moreover, the serum half-life of the heterodimer was significantly longer (Fig. 2i). Heterodimer exhibited more bioactivity stability than homodimer, when incubated in the

serum at 37℃ in vitro (Supplementary Fig. 5a, b). These data were consistent with the better antitumor effects of the heterodimer, and suggested that the heterodimer format is a better candidate for in vivo study. Taken together, these data show that targeted delivery of IFNα by anti-PD-L1 induces potent antitumor effects, leading to improved tumor control.

**IFNα-anti-PD-L1 retains in tumor reducing systemic toxicity.** The clinical utilization of type I IFNs has been limited due to severe side effects when delivered systemically[19,24]. It is hard to evaluate the immune-related adverse effect within murine model, because mice are usually less sensitive to the drug. To test the in vivo toxicity of IFNα-armed anti-PD-L1, mice were treated with either a high-dose of the heterodimer (IFNα-anti-PD-L1, thereafter) or a non-targeting control IFNα-anti-HBs (IFNα-armed anti-hepatitis B virus surface protein) fusion protein. IFNα-anti-PD-L1 and IFNα-anti-HBs had similar in vitro bioactivity (Supplementary Fig. 6a). After the second challenge of IFNα-anti-HBs, tumor-bearing mice experienced severe weight loss, reduced mobility, ruffled fur, and all died within 1 day (Fig. 3a). In contrast, none of the mice treated with IFNα-anti-PD-L1 died, recovering after mild weight loss (Fig. 3a, b). To better evaluate the side effects of fusion proteins, cytokine levels in the serum were measured after the first injection. Non-targeting IFNα-anti-HBs induced an impressive cytokine storm with high levels of TNF, IFNγ, MCP-1, IL-6, and IL-10 (Fig. 2c and Supplementary Fig. 6b). The IFNα-anti-PD-L1 only induced bearable cytokines release in the tumor-bearing mice. The in vivo biodistribution of the IFNα-anti-PD-L1 fusion protein also displayed considerable specificity; high concentration of the fusion protein were retained within the tumor even to day 5, and much less were found in normal tissues (Fig. 3d). Further bioluminescence data showed that IFNα-anti-PD-L1 fusion protein did accumulate within tumor tissue after injection (Fig. 3e). This tumor-specific targeting is critical for the antitumor effects, as a

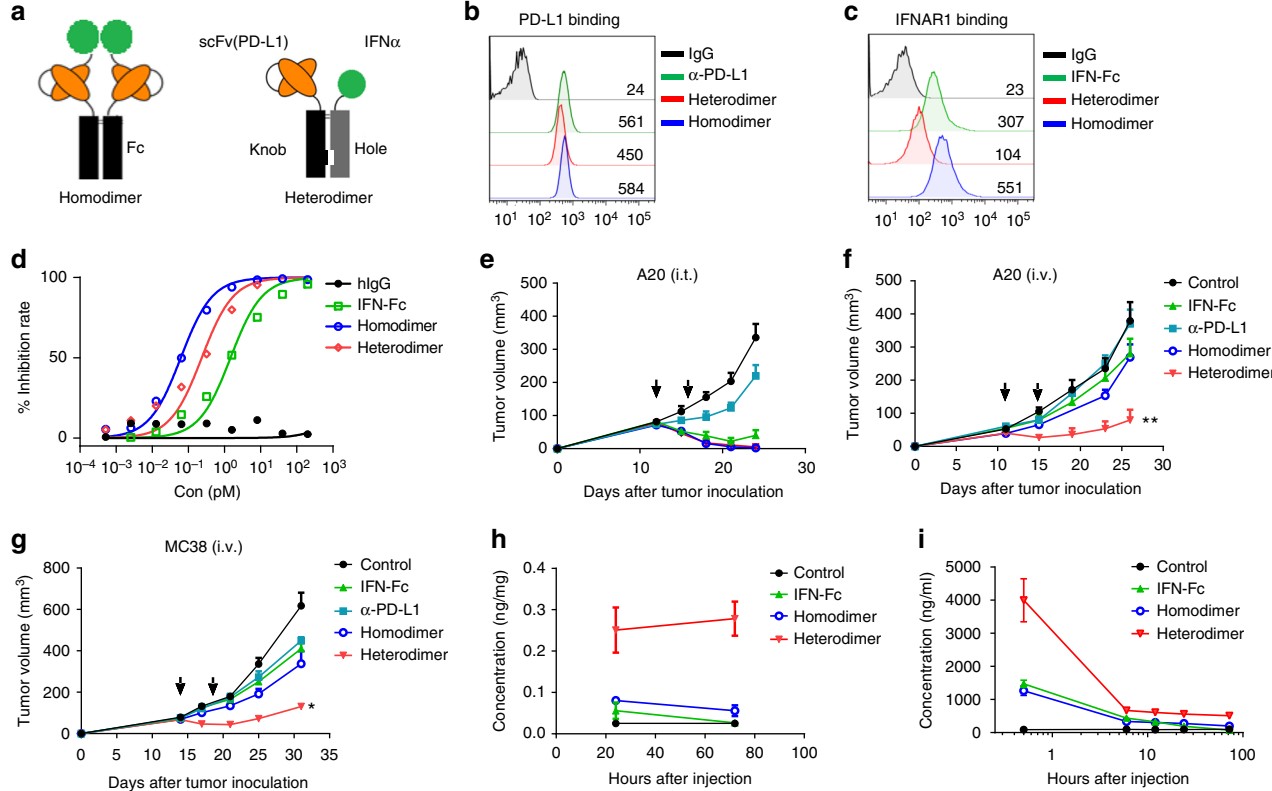

**Fig. 2** Construction and characterization of IFNα-armed anti-PD-L1. **a** Schematic representations of IFNα-anti-PD-L1 fusion protein in homodimer or heterodimer format. scFv, single-chain variable fragment. **b–c** Flow cytometry histograms showing the binding of indicated protein in IFNAR1$^{-/-}$ A20 cells (**b**) and PD-L1$^{-/-}$ A20 cells (**c**) at the concentration of 80 nM. Numbers indicate the mean fluorescent intensity (MFI). **d** The bioactivity of IFNα-anti-PD-L1 protein was measured by an anti-viral infection biological assay. L929 cells were cultured with each protein overnight before being infected with VSV-GFP virus. After another 30 h of culture, the percentage of virus-infected cells was determined by flow cytometry. **e, f** Balb/c mice ($n = 5$) were inoculated with $3 \times 10^6$ A20 cells. After tumor established, mice were treated with 20 µg of control, anti-PD-L1, IFNα-Fc, or fusion protein by i.t. (**e**) or i.v. (**f**, day 11 and 15) injection. Tumor size was measured twice per week. **g** C57BL/6 mice ($n = 4$–8) were inoculated with $5 \times 10^5$ MC38 cells. Mice were treated i.v. with 25 µg control or fusion protein on days 14 and 18. **h, i** Mice were injected i.v. with 25 µg indicated protein. Protein concentrations in tumor tissues (**h**) or serum (**i**) at different time points were measured by ELISA. Data indicate mean ± SEM and are representative of at least two independent experiments. *$p < 0.05$; **$p < 0.01$

simple mixture of IFNα-Fc with anti-PD-L1 did not produce the synergistic effects that the IFNα-anti-PD-L1 fusion protein did (Fig. 3f).

Since type I IFNs is one of the most potent cytokines that induce PD-L1 expression[14,15], we measured PD-L1 levels in the TME after systemic treatment with IFNα-anti-PD-L1. IFNα-anti-PD-L1 was able to significantly increase PD-L1 expression within the TME in both WT and T-cell deficient Rag-1 mice (Fig. 3g and Supplementary Fig. 7), suggesting that the induction of PD-L1 does not depend on activation of T cells. Increased PD-L1 expression may enhance tumor-specific accumulation of the fusion protein. Taken together, our data show that IFNα-armed with anti-PD-L1 creates feedforward response, which may further enhance tumor-targeting, and control tumor growth with minimal toxicity.

**PD-L1 in tumor cells is dispensable for IFNα recruitment**. Many tumor cells overexpress PD-L1 as a strategy to evade immune responses[13]. PD-L1 can also be induced by inflammatory cytokines in many cells besides tumor cells[25]. There is intense debate concerning whether PD-L1 in tumor cells or non-tumor cells correlates better with PD-L1 blockade therapy[26–29]. In this study, we explored the role of PD-L1 in the tumor cells or host cells during the IFNα-anti-PD-L1 treatment. To determine if

PD-L1 in tumor cells is essential, we knocked out PD-L1 using CRISPR/Cas9 technology. PD-L1 expression was found abolished in knockout A20 and MC38 tumor established in WT mice (Fig. 4a). When stimulated by IFNα, wild-type (WT) cells upregulated PD-L1 while knockout cells remained negative, indicating a complete ablation of PD-L1 expression (Supplementary Fig. 8). WT and PD-L1-knockout tumor-bearing mice were treated with fusion protein to test if PD-L1 on tumor cells is essential for the targeting effects. The levels of protein in the tumor tissues were measured. To our surprise, the fusion protein accumulated in tumor tissues regardless of the status of tumor PD-L1 expression (Fig. 4b). PD-L1-knockout tumors treated with fusion protein could be controlled as effectively as WT tumors (Fig. 4c, d). These data suggest that PD-L1 in tumor cells are dispensable for yielding antitumor effects.

This study showed that the IFNα-anti-PD-L1 created a feedforward response to upregulate PD-L1 expression in the TME (Fig. 3g). Since both tumor and stromal cells can express PD-L1, we measured their respective PD-L1 expressions. IFNα-anti-PD-L1 treatment dramatically upregulated PD-L1 expression in both tumor and stromal cells (Fig. 4e). Since PD-L1 was nonessential in tumor cells, we wondered if PD-L1 is required in host cells. Interestingly, IFNα-anti-PD-L1 was able to control tumor growth in PD-L1-deficient mice (Fig. 4f). In summary, these data suggest that PD-L1 expressed in either host or tumor

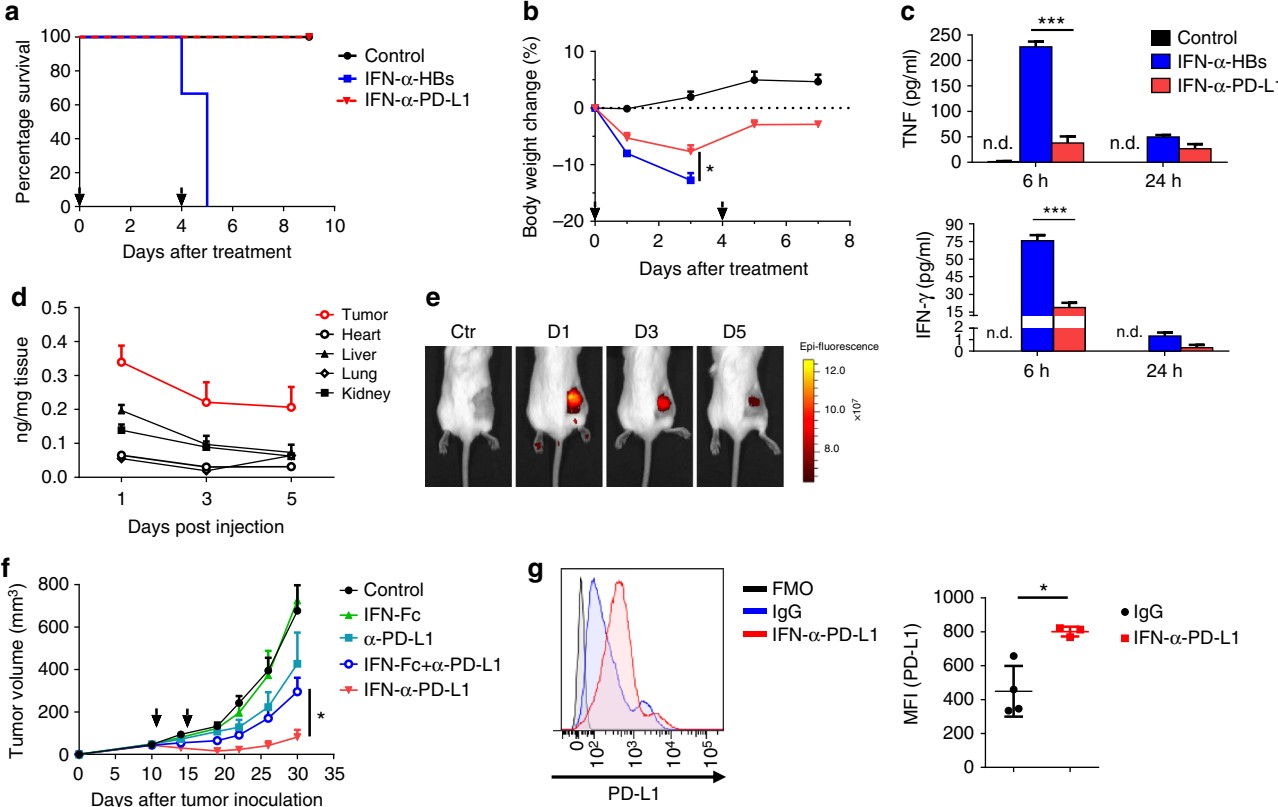

**Fig. 3** IFNα-armed anti-PD-L1 leads to maximal antitumor activities with minimal toxicities. **a**, **b** MC38 tumor-bearing mice were injected i.v. with 100 μg IFNα-anti-PD-L1 heterodimer or IFNα-anti-HBs protein on day 0 and 4. The mice survival curves (**a**) and body weight changes (**b**) were shown. **c** MC38 tumor-bearing mice were injected i.v. with 100 μg IFNα-anti-PD-L1 or IFNα-anti-HBs protein. Serum was collected at 6 or 24 h after injection. Cytokine levels in the serum were measured by cytometric bead array (CBA). **d** MC38 tumor-bearing mice were injected i.v. with 30 μg of IFNα-anti-PD-L1. Tissues were collected on days 1, 3, and 5 after injection. Concentrations of fusion protein were measured by ELISA. **e** IFNα-anti-PD-L1 protein was conjugated with Cy5.5-maleimide. 25 μg of CY5.5- IFNα-anti-PD-L1 protein was injected into A20 tumor-bearing mice. The protein accumulating within tumor was tracked by IVIS spectrum in vivo imaging system. **f** A20 tumor-bearing mice ($n = 5$) were treated i.v. with IFNα-Fc (12.5 μg), anti-PD-L1 (12.5 μg), a mixture of IFNα-Fc and anti-PD-L1 (12.5 μg + 12.5 μg), or IFNα-anti-PD-L1 fusion protein (25 μg) on days 11 and 15. Tumor size was measured twice a week. **g** MC38 tumor-bearing mice were treated with 25 μg control Ig or IFNα-anti-PD-L1. Two days later, tumor tissues were harvested and PD-L1 levels were determined by flow cytometry. A representative graph is shown on the left, and MFI is shown on the right. FMO fluorescence minus one. Data indicate mean ± SEM and are representative of at least two independent experiments. *$p < 0.05$; **$p < 0.01$; ***$p < 0.001$; n.d. not detectable, n.s. not significant

cells is sufficient for both the tumor-targeting and antitumor effects of IFNα-anti-PD-L1.

**Activation of IFNAR in the host cells mediates tumor control.**
Next, we investigated whether type I IFN signaling is necessary for antitumor effects by manipulating the IFNAR, which is expressed in both tumor and host cells. Mice were treated with anti-IFNAR blocking antibody during fusion protein treatment, which resulted in completely abrogated the antitumor effects, suggesting an essential role of type I IFN signaling (Fig. 5a). In order to examine whether IFN-receptor in tumor cells is required for IFNα induced tumor killing, we knocked out IFNAR1 (Fig. 5b), and found that IFNα-anti-PD-L1 continued to efficiently control tumor growth in mice bearing IFNAR1$^{-/-}$ A20 tumors (Fig. 5c). Since the IFN-receptor in tumor cells is thus not essential, we then investigated whether receptor expressed on host cells is essential. MC38 tumors were inoculated into WT or IFNAR$^{-/-}$ mice and treated with fusion protein. The antitumor effects disappeared in IFNAR$^{-/-}$ mice, suggesting that the receptor is, in fact, essential in the host cells (Fig. 5d). IFNα act as a third signal during T-cell priming, could directly promote the proliferation and increase the cytokine secretion[30,31]. CD8$^+$ T cells are critical for the antitumor effects, as depletion with anti-CD8 antibody completely abrogated the effects

(Fig. 5e). Both IFNα-Fc and IFNα-anti-PD-L1 fusion proteins could indeed activate murine CD8$^+$ T cells efficiently in vitro (Supplementary Fig. 9a). On the basis of these results, we investigated the necessity of IFNAR signaling in CD8$^+$ T cells during IFNα-anti-PD-L1 treatment. We transferred sorted WT or IFNAR$^{-/-}$ CD8$^+$ T cells into tumor-bearing Rag-1 mice and treated them with IFNα-anti-PD-L1 protein. IFNα-anti-PD-L1 continued to inhibit tumor growth when IFNAR in CD8$^+$ T cells was deficient (Supplementary Fig. 9b). This is consistent with our previous data that type I IFN could directly engage IFNAR in T cells for an improved antitumor effect, but it was the type I IFN targeted DC cross-presentation that mainly contributed to better T-cell activation[19]. As predicted, CD86 and CD80 FACS staining results revealed that IFNα-anti-PD-L1 treatment increased DC activation (Fig. 5f and Supplementary Fig. 9c). In the control, no significant activation of DC was observed in IFNα-Fc-treated tumors, suggesting the importance of its tumor-specific targeting ability. Altogether, these data demonstrate that IFNα-anti-PD-L1 fusion protein mediates its antitumor effects mainly through type I IFN signaling in host cells.

**IFNα-anti-PD-L1 reverses tumor resistance to PD-1 blockade.**
Advanced tumors were often resistant to PD-L1 blockade therapy. In fact, neither anti-PD-1 nor anti-PD-L1 treatment could

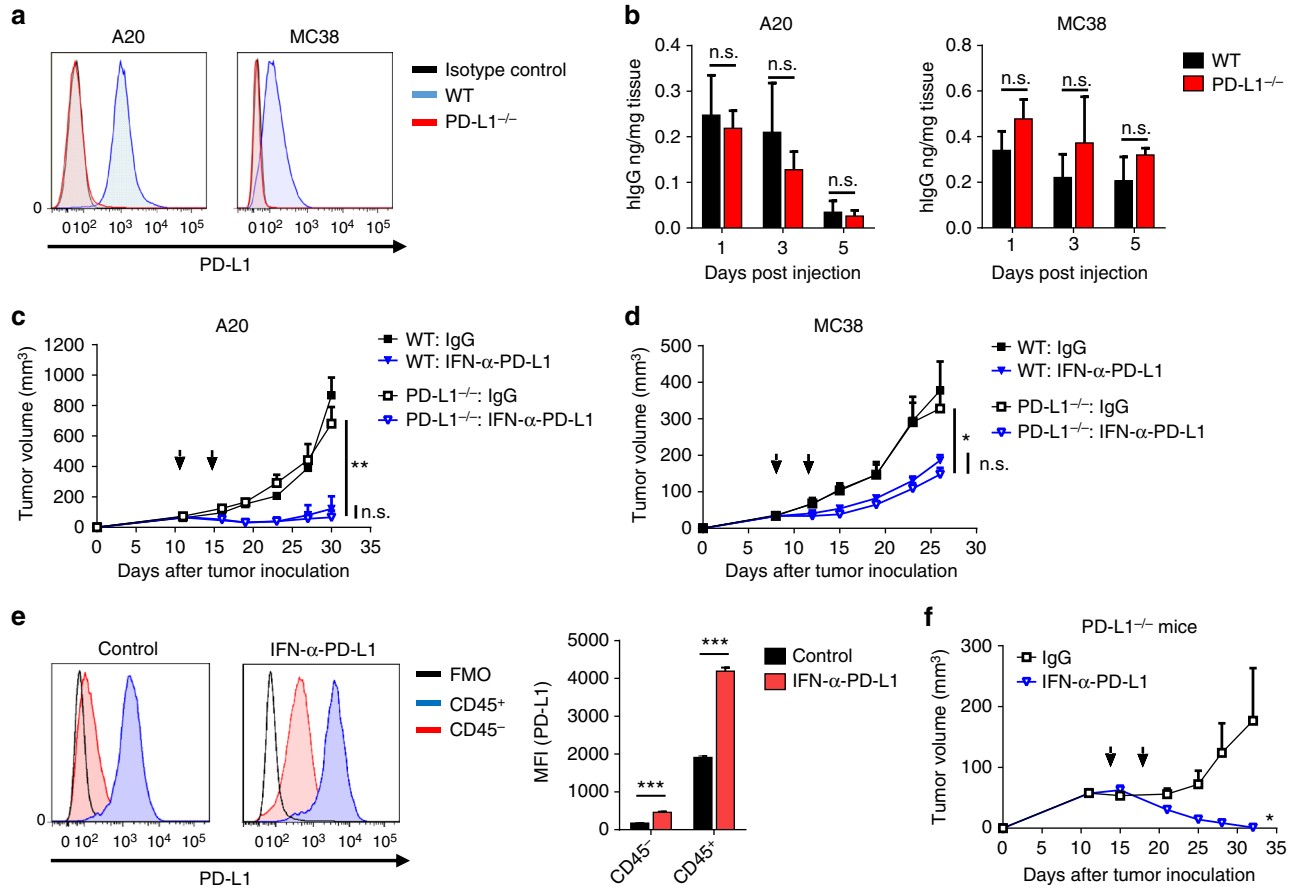

**Fig. 4** PD-L1 expressed in either host or tumor cells is sufficient for the antitumor effects of IFNα-armed anti-PD-L1. **a** PD-L1 levels in B220+ cells of WT and PD-L1−/− A20 tumors, and CD45− cells of WT and PD-L1−/− MC38 tumors were measured by flow cytometry. **b** WT or PD-L1−/− tumor-bearing mice were injected i.v. with 30 μg of IFNα-anti-PD-L1. Tumor tissues were collected at various time points post injection. Concentrations of fusion protein were measured by ELISA. **c** Mice (*n* = 4–5) bearing WT or PD-L1−/− A20 tumors were treated with control Ig or IFNα-anti-PD-L1 on days 11 and 15. Tumor growth was measured twice per week. **d** Mice (*n* = 5–6) bearing WT or PD-L1−/− MC38 tumors were treated with control Ig or IFNα-anti-PD-L1 on days 8 and 12. **e** MC38 tumor-bearing mice were treated with IFNα-anti-PD-L1. Two days later, tumor tissues were collected. PD-L1 levels in CD45− vs. CD45+ cells were evaluated by flow cytometry. A representative graph is shown on the left, and MFI is shown on the right. **f** PD-L1−/− mice (*n* = 4–5) were inoculated with MC38 cells. Mice were treated with 25 μg of control Ig or IFNα-anti-PD-L1 on days 14 and 18. Tumor growth was measured twice per week. Data indicate mean ± SEM and are representative of at least two independent experiments. *$p < 0.05$; **$p < 0.01$; ***$p < 0.001$; n.s. not significant

control the growth of advanced A20 tumors (Fig. 6a). In comparison to the PD-1 blockade therapy, the IFNα-anti-PD-L1 fusion protein showed better antitumor effects. However, some tumors eventually relapsed after initial control (Fig. 2f, g). It is possible that excessive PD-L1 expression after immune treatment may further limit T-cell-mediated tumor control. Therefore, we hypothesized that a combination therapy of anti-PD-1 and IFNα-anti-PD-L1 might be able to overcome tumor resistance to either IFNα or PD-L1 blockade therapy. Combination therapy led to much better tumor control, and most advanced tumors completely regressed (Fig. 6a). B16 is a well-known mouse tumor model that is resistant to PD-1 blockade therapy. Consistent with previous reports, PD-1 blockade therapy had no effects on tumor growth in the B16 model (Fig. 6b). IFNα-anti-PD-L1 treatment could only partially control the tumor growth. Interestingly, combination therapy of IFNα-anti-PD-L1 treatment with PD-1 blockade dramatically improved the antitumor effects.

Since it would be clinically valuable for a patient to eventually be able to prevent the tumor relapse, we determined if the combination therapy of IFNα-anti-PD-L1 with PD-1 blockade-mediated antitumor responses result in prolonged protective immunity. Mice that had already undergone complete tumor regression after combinational therapy were re-challenged with a

lethal dose of A20 cells. All the mice rejected the re-challenged tumor, confirming a strong memory response (Fig. 6c). To determine which cell population(s) is necessary for the combination therapy, mice that underwent combination therapy were depleted of NK, CD4+, or CD8+ T cells via antibodies. In the absence of CD8+ T cells, the antitumor effects completely disappeared (Fig. 6d). In contrast, depletion of NK or CD4+ T cells had limited effects (Supplementary Fig. 10a, b). We also want to establish whether or not tumor-specific T cells increased as a result of treatment; cells were thus isolated from lymph node or spleen tissues and co-cultured with irradiated A20 tumor cells. An IFN-γ ELISPOT assay was performed to evaluate tumor-specific T-cell responses. While PD-1 blockade therapy alone had limited effects on T-cell activation, IFNα-anti-PD-L1 induced slightly better responses (Fig. 6e and Supplementary Fig. 10c). Most importantly, combination therapy dramatically increased the number of tumor-specific T cells (Fig. 6e and Supplementary Fig. 10c). In addition, we tested the effects of IFNα, PD-L1, and tumor cells in a defined in vitro system that could better recapitulate the TME. DCs and T cells were isolated from the in vivo established tumor and co-cultured with tumor cells for 3 days in the presence of IFNα, anti-PD-L1, or a combination of both. While single treatment had limited effects, combination

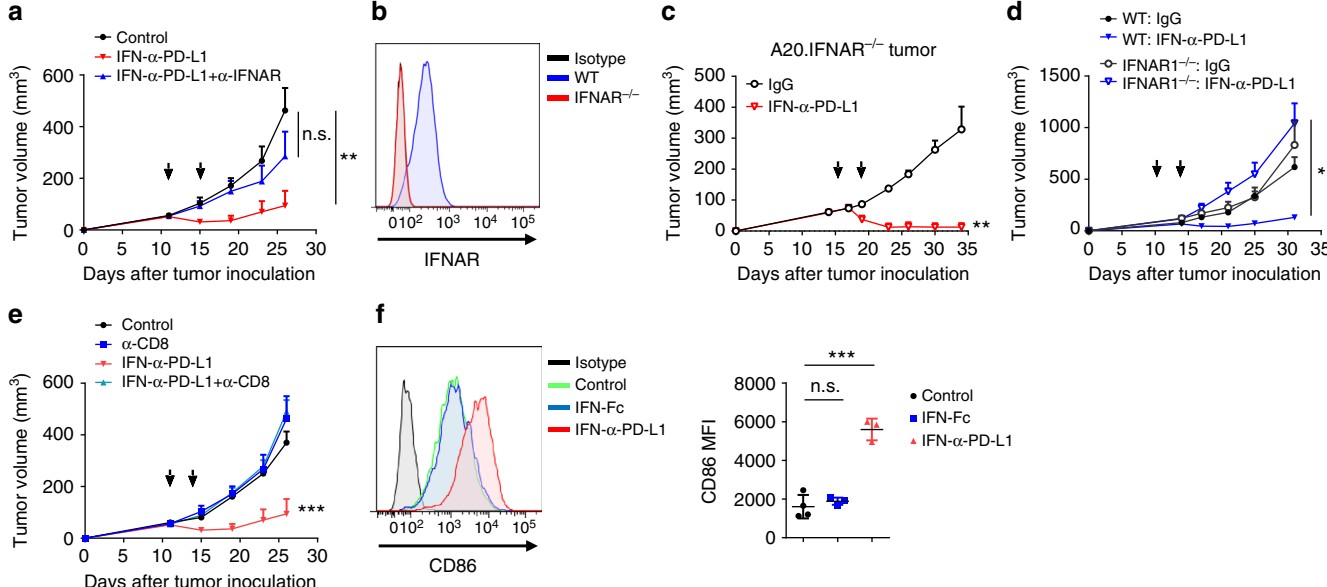

**Fig. 5** IFNAR signaling in the host is essential for tumor control. **a** A20 tumor-bearing mice ($n = 5$) were treated with IFNα-anti-PD-L1 on day 11. To block type I IFN signaling, 100 μg of anti-IFNAR blocking antibody was injected i.t. into mice on days 11 and 14. **b** IFNAR expression in WT or IFNAR$^{-/-}$ A20 tumor cells in vivo (B220$^+$) was evaluated by flow cytometry. **c** IFNAR$^{-/-}$ A20 tumor-bearing mice ($n = 6$) were treated with IFNα-anti-PD-L1 on days 16 and 19. Tumor growth was measured twice per week. **d** WT or IFNAR1$^{-/-}$ mice were inoculated with $5 \times 10^5$ MC38 cells. Mice ($n = 4$–5) were treated with 25 μg of IFNα-anti-PD-L1 on days 10 and 13. Tumor growth was measured twice per week. **e** MC38 tumor-bearing mice ($n = 5$–6) were treated with IFNα-anti-PD-L1 on days 11 and 14. Anti-CD8 depletion antibody was administered on days 9, 12, and 16. Tumor growth was measured twice per week. **f** Two days after IFNα-anti-PD-L1 treatment, MC38 tumor tissues were isolated. Expression of CD86 in tumor-infiltrating DCs (CD11c$^+$MHCII$^+$) was measured by flow cytometry. A representative graph is shown on the left, and MFI is shown on the right. Data indicate mean ± SEM and are representative of at least two independent experiments. $*p < 0.05$; $**p < 0.01$; $***p < 0.001$; n.s. not significant

treatment of IFNα and anti-PD-L1 significantly increased IFNγ production by T cells (Fig. 6f). These data suggest that combining IFNα and PD-1/PD-L1 blockade therapy coordinately induces a robust tumor-specific T-cell response that can overcome tumor resistance to checkpoint blockade in advanced tumors.

**IFNα-anti-PD-L1 activates resident T cells for tumor control.** Our data show that tumor-specific T cells play essential roles in antitumor immune responses. These T cells can potentially originate from two major sources: pre-existing T cells inside tumor tissues, or newly activated T cells migrating from peripheral lymphoid tissues into tumor. To test which T-cell populations are essential, we either utilized FTY720 to block peripheral lymphocytes from trafficking into tumor tissues or used an anti-CD8 antibody to deplete CD8$^+$ T. Here, we found that combination therapy of IFNα and PD-1 blockade controlled the FTY720 administrated tumor similarly to that of the control group, while local depletion of CD8$^+$ T cells diminished all the antitumor effects (Fig. 6g). These data suggest that reactivation of pre-existing T cells by IFNα-anti-PD-L1 is sufficient for tumor control. Patients who underwent PD blockade therapy with higher IFNA1 expression within tumor tissues positively correlated with better prognosis, which is consistent with this study (Fig. 6h). Taken together, these data suggest that PD-1/PD-L1 blockade can rescue T cells from an exhausted status, while IFNα effectively reactivates partially recovered resident T cells for tumor control.

## Discussion

Although PD-1/PD-L1 signaling is an important mechanism utilized by tumors to evade antitumor immune responses, objective responses are only observed in a minority of patients after PD-1/PD-L1 blockade therapy[2,3]. Many patients, especially those with advanced tumors, do not respond to this therapy

despite the presence of heavy lymphocyte infiltration in tumors[6]. Even in responding patients, resistance to PD-1/PD-L1 blockade therapy may develop after prolonged treatment[1,4]. In this study, we showed that PD-L1 blockade therapy controls early-stage but not advanced tumors in mouse models. We observed that exogenous type I IFN in the TME helps to provoke antitumor T-cell responses, leading to better tumor control. Type I IFNs can promote DC cross-presentation and T-cell activation. However, it could upregulate inhibitory PD-L1 molecule, which dampen the subsequent T-cell response. We proposed that anti-PD-L1 antibody can not only block PD-L1 signaling but also deliver IFNα specifically into tumor tissues. To test this concept, we generated a next-generation anti-PD-L1 antibody armed with IFNα (Supplementary Fig. 11). In different tumor models, we have shown that IFNα-anti-PD-L1 could control advanced tumors better than anti-PD-L1. The antitumor effects depended on CD8$^+$ T cells, but not NK or CD4$^+$ T cells. Interestingly, PD-L1 expressed on either tumor or host cells was sufficient for the antitumor effects. Moreover, IFN-receptor signaling on host cells but not tumor cells was necessary. When combined with PD-1 blockade, IFNα-anti-PD-L1 completely eradicated most PD-1 blockade-resistant tumors and provide long-term immunological memory. Therefore, we have developed a next-generation anti-PD-L1 antibody that is able to overcome innate and adaptive tumor resistance to checkpoint blockade therapy. Consistent with our conclusions, patients that have higher type I IFN levels in tumor tissues not only showed better prognosis, but also responded better to PD-1/PD-L1 blockade therapy.

Type I IFNs have been approved by the FDA for the treatment of melanoma and lymphoma. Recently, we and others have found that endogenous type I IFNs play a critical role in various antitumor therapies[9,11,12]. It has been suggested that the delivery of exogenous type I IFN may work as a "magic bullet" that impacts

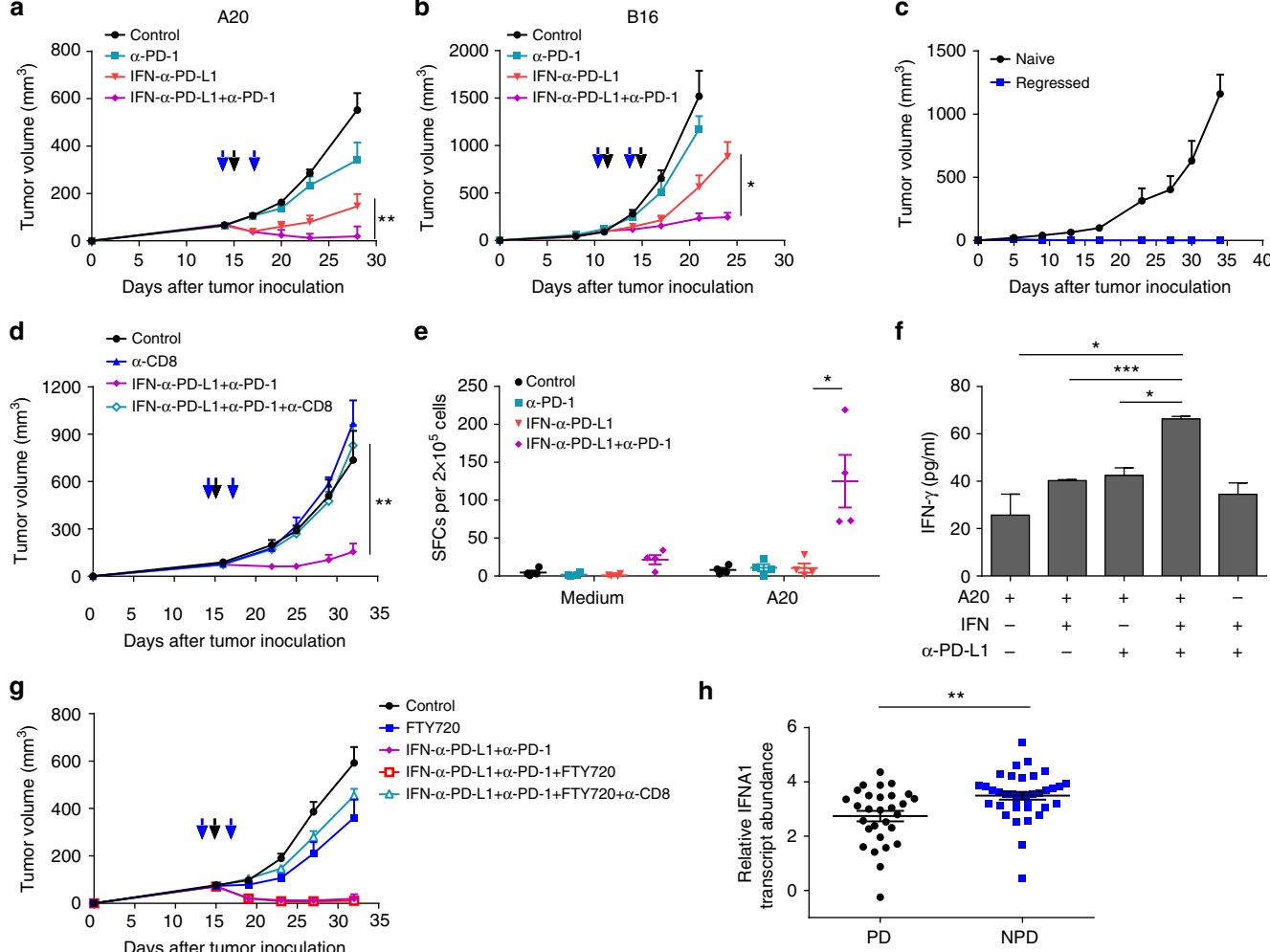

**Fig. 6** PD-1 blockade further ensures that IFNα-anti-PD-L1 induces feedforward antitumor responses. **a** A20 tumor-bearing Balb/c mice (*n* = 4–5) were treated with 20 μg of IFNα-anti-PD-L1 on day 15 and/or 100 μg anti-PD-1 on days 14 and 17. Tumor growth was measured twice per week. **b** B16 tumor-bearing C57BL/6 mice (*n* = 3–5) were treated with 25 μg of IFNα-anti-PD-L1 on days 11 and 14 and/or 100 μg of anti-PD-1 on days 12 and 15. Tumor growth was measured twice per week. **c** Mice (*n* = 4) with complete regression after combination therapy in **a** were re-challenged with 2.5 × 10⁷ A20 cells. Naive mice inoculated with A20 cells were used as a control. Tumor growth was measured twice per week. **d** Mice (*n* = 4) were treated with IFNα-anti-PD-L1 and anti-PD-1 as in **a**. For cell depletion, mice were injected with 200 μg of anti-CD8 antibody from 1 day before anti-PD-1 treatment. Tumor growth was measured twice per week. **e** Mice were treated with IFNα-anti-PD-L1 and/or anti-PD-1 as in **a**. Twelve days after treatment, tumor-draining LNs were isolated and single-cell suspensions were prepared. Cells were co-cultured with or without irradiated A20. The IFNγ ELISPOT assay was performed. **f** Tumor-infiltrating DC (CD11c⁺) and T (CD8⁺) cells were isolated from A20 tumor-bearing mice and co-cultured in the presence of irradiated A20 cells. IFNα-Fc or anti-PD-L1 was added to the culture medium. Three days later, supernatants were harvested and IFNγ levels were measured by CBA. **g** A20 tumor-bearing mice (*n* = 5–6) were treated with IFNα-anti-PD-L1 and/or anti-PD-1 as in **a**. FTY720 was administered every other day from day 14. To deplete intratumoral CD8⁺ T cells, 30 μg anti-CD8 antibody was injected i.t. on days 14 and 17. Tumor growth was measured twice per week. Blue and black arrows indicate treatment with anti-PD-1 and IFNα-anti-PD-L1, respectively. **h** Patients treated with anti-PD-1 antibody were grouped as progressive disease (PD, *n* = 29) or non-progressive disease (NPD, *n* = 36). Relative IFNA1 transcript abundances were compared. Data indicate mean ± SEM and are representative of at least two independent experiments. *$p < 0.05$; **$p < 0.01$; ***$p < 0.001$

both immune responses and tumor cell proliferation/survival[24]. However, high-dose of type I IFN need to be administered systemically to achieve high-serum concentration for effective clinical therapeutic effect. In this situation, dose-limiting toxicities are found and restrict its application. In addition, IFN is one of the most potent inducers of PD-L1, which serves as a negative feedback mechanism to prevent tissue damage but also dampens the antitumor effects[13,32]. Since PD-L1 is highly expressed in many tumors but not normal tissue, we utilized anti-PD-L1 to specifically introduce IFNα into tumor tissues. The therapeutic effect of IFNα-anti-PD-L1 was much better than the mixture of anti-PD-L1 and IFNα-Fc, which suggests the importance of IFNα tumor-targeting. IFNα upregulates PD-L1 expression in the TME

directly. Such a feedforward loop can further enhance the targeting effect. In addition, IFNα-anti-PD-L1 activates IFN receptors in immune cells within tumor tissues. These feedforward responses lead to maximal antitumor activities with the least side effects. It has been reported that targeted activation of type I IFN axis by peri-tumoral injection of poly (I:C) induces antitumor immune responses, and the effects can be further enhanced by a combination therapy with PD-1 blockade. Comparing to their study, our design has several advantages. First, peri-tumoral injection is not feasible to most patients in clinic. Our design that utilizing anti-PD-L1 as a carrier to deliver IFNα into tumor tissues overcome the targeting problem. Second, our fusion protein has significantly reduced the side effects due to the

tumor-targeting effects. Third, IFNα-anti-PD-L1 as a protein drug is easier to be accepted than poly (I:C) clinically.

Meanwhile, our study suggest that it is important to choose the right format for the construction of cytokine-armed antibodies. The homodimer is one of the most common-used format for antibody-cytokine fusion design. Although the homodimer showed higher IFN-receptor-binding affinity and more potent anti-viral activity in vitro, the heterodimer has much better tumor-targeting, longer serum half-life, and better tumor control in vivo (Fig. 2). Interestingly, a recent study showed that decreasing IFN-receptor-binding affinity through mutation enhances tumor-targeting of IFNα, probably due to reduced sink effect in the peripheral, which is consistent with the results of this study[29].

Many tumors overexpress PD-L1 as a strategy to evade anti-tumor immune responses. In contrast, PD-L1 expression in normal tissues is rare[13,32,33]. The unique expression of PD-L1 makes it an interesting antigen for tumor-targeting. In fact, we have observed a specific accumulation of anti-PD-L1 antibodies in tumor tissues in mice[23]. Here, we demonstrated that anti-PD-L1 can be utilized to deliver immunomodulatory molecules specifically into tumor tissues. These findings lay the foundation for the development of anti-PD-L1 for tumor-targeting. However, different tumors express various levels of PD-L1. In some tumors, the PD-L1 levels may not be sufficient for effective antibody targeting. Intriguingly, our data show that PD-L1 expressed in either tumor or host cells is sufficient for the targeting and antitumor effects. The advantages of IFNα-anti-PD-L1 is that it forms a feedforward loop to enhance targeting effects by upregulating PD-L1 expression, which is superior for treating tumors with a lower level of PD-L1.

Our study reveals several mechanisms that could be used for further cancer immunotherapy. First, our data suggest that activation of IFN signaling in the TME can potentiate PD-L1 blockade therapy against advanced tumors by inducing more robust T-cell activation. Second, by generating an IFNα-armed anti-PD-L1, we show that anti-PD-L1 antibody can be utilized to deliver immunomodulatory molecules specifically into tumor tissues with the least toxicity. Our study lays the foundation for developing anti-PD-L1 for tumor-targeting. Third, IFNα-armed anti-PD-L1 creates multiple feedforward response that increase targeting effects to enhance responses to IFNα treatment, thereby maximizing antitumor effects. Although IFNα-anti-PD-L1 can increase PD-L1 expression for better targeting, it remains to be determined whether PD-L1 expression before treatment is necessary for the initial accumulation of IFNα-PD-L1 in tumor tissues. Nevertheless, our data suggest that the next-generation PD-L1 antibody armed with IFNα can overcome tumor resistance to checkpoint blockade therapy in advanced tumors. The combination therapy of type I IFN with PD-1 blockade Ab for patients with advanced melanoma tested in phase I study have demonstrated the safety, and several other clinical trials are ongoing[34]. Our strategy of constructing IFNα-anti-PD-L1 may be important for improving current immunotherapies.

## Methods

**Mice**. Female (6–8-weeks-old) BALB/c and C57BL/6J mice were purchased from Vital River Laboratories (Beijing, China). All mice were maintained under specific pathogen-free (SPF) conditions in the animal facility of the Institute of Biophysics. Animal care and experiments were performed in accordance with the guidelines of the Institute of Biophysics, Chinese Academy of Sciences, using protocols approved by the Institutional Laboratory Animal Care and Use Committee. PD-L1[-/-] and IFNAR1[-/-] mice were maintained under SPF conditions at UT Southwestern Medical Center. Animal protocols were consistent with National Institutes of Health guidelines. Studies were approved by the Animal Care and Use Committee of UT Southwestern Medical Center.

**Cell lines and reagents**. The 293F cells were kindly provided by Dr. Ting Xu (Alphamab, Suzhou, Jiangsu, China) and grown in SMM 293-TI medium (M293TI, Sino Biological). A20, MC38, B16F10, and L929 cell lines were purchased from ATCC (Manassas, VA). PD-L1-deficient MC38 and A20 cell lines were generated by CRISPR/Cas9 technology. The guide sequences (5′-GACTTGTACGTGGTGGA GTA-3′) and EGFP gene were cloned into lentiCRISPR v2 plasmid (Addgene, catalog 52961). Thirty hours post transfection, EGFP-positive cells were sorted and subcloned by flow cytometry. Identified through FACS staining, cell clones without PD-L1 or EGFP expression were used for the following studies.

Anti-PD-1 blockade Ab (4H2) was from Bristol-Myers Squibb (Redwood City, CA). Anti-PD-L1 (10F.9G2) and anti-IFNAR1 (MAR1–5A3) antibodies were purchased from BioXCell (West Lebanon, NH). Anti-CD8 (TIB210) and anti-CD4 depleting Ab (GK1.5) were produced in house. Anti-Asialo-GM1 antibody were purchased from Biolegend (San Diego, CA).

**Production of IFNα-anti-PD-L1 fusion protein**. The variable regions of the light-chain and heavy-chain of the anti-PD-L1 Ab (YW243.55.S70) sequence were synthesized according to a patent (Patent No.: US8217149 B2). The light-chain and heavy-chain sequences were joined with a GGGGSGGGGSGGGGSGGGGS linker, and human IgG1 Fc was inserted into the C-terminus of the heavy chain. The entire sequence was then cloned into the pEE12.4 vector (Lonza). Murine IFNα4 cDNA sequence was cloned and inserted into the N terminal of human IgG Fc. IFNα with Fc was cloned into the pEE6.4 vector (Lonza). Heterodimerization of anti-PD-L1 and IFNα was achieved by using the knobs-into-holes technique as previously reported[35]. The variable region of anti-PD-L1 was fused to the "Knobs" (T366W) arm, and IFNα4 was fused to the "Holes" (T366S, L368A, Y407V) arm (Fc CH2 CH3). The anti-PD-L1-knobs and IFNα4-hole plasmids were transiently transfected into 293F cells at a ratio of 1:2. Supernatants were collected on day 7 after transfection. The fusion protein was purified using a Protein A-Sepharose column according to the manual (Repligen Corporation). All the proteins have a yield of 50–100 mg/l before purification. The harvest efficiency of proteins is about 60% through purification. For in vivo imaging, IFNα-anti-PD-L1 protein was conjugated with a near-IR fluorescence dye, Cy5.5−maleimide, with a molar ratio of 1:10 in PBS buffer (pH 7.4) at 4 ℃ overnight. The excess of Cy5.5 was removed by desalting column.

**Binding assay of IFNα-anti-PD-L1 to PD-L1**. The affinities of proteins for PD-L1 binding were measured using biolayer inferometry (BLI) on an Octet Red96 system (ForteBio). Binding experiments were performed at 37 ℃ in binding buffer (PBS, 0.05% Tween 20, pH 7.4). Seven SA sensors were used to load Biotinylated PD-L1 protein in the 96-well. The sensors were moved to binding wells for baseline generation and followed to sample proteins wells (concentration range is 30 nM, 15 nM, 7.5 nM, 3.75 nM, 1.88 nM, and 0.94 nM) for 120 s. Then they were dipped to binding buffer wells for dissociation for 300 s. The constants were determined by dynamic-analysis model. $K_D = k_d/k_a$ (d, dissociation; a, association).

**Flow cytometry**. Binding of the fusion protein was detected using PE-anti-human IgG Fc (clone HP6017,1 μg/ml). PD-L1 (clone 10 F.9G2, 1 μg/ml), IFNAR1 (clone MAR1-5A3, 1 μg/ml), CD45 (clone 30-F11, 0.5 μg/ml), CD80 (clone 16-10A1, 0.5 μg/ml), CD86 (clone GL1, 0.5 μg/ml) were from BioLegend or eBioscience. Cells suspended in FACS buffer (1% bovine serum albumin and 0.05% NaN3) were blocked with anti-CD16/32 (anti-FcγIII/ II receptor, clone 2.4G2) for 30 min and then stained with specific antibodies for 30 min on ice. Samples were analyzed on a FACS Calibur or Fortessa flow cytometer (BD Biosciences). Data were analyzed using FlowJo software (TreeStar).

**IFNα anti-viral activity**. The L929 fibroblast cell line sensitive to VSV infection was used to quantify the biological activity of IFNα. Cells were incubated with serial dilutions of IFNα-Fc or IFNα-anti-PD-L1 at 37 ℃ overnight. The next day, the cells were infected with VSV-GFP at MOI = 5 and cultured for another 30 h. Cells were then collected and fixed with 4% PFA. Data were acquired using a FACS Fotassa flow cytometer (BD Biosciences) and analyzed using FlowJo software (TreeStar). GFP-positive cells were defined as virus-infected cells.

**Quantitative biodistribution studies**. A20 cells ($3 \times 10^6$) were injected subcutaneously (s.c.) into the right flank of Balb/c mice. Mice were treated intravenously with 30 μg IFNα-Fc or IFNα-anti-PD-L1 on day 15. Three days later, different mouse tissues were collected after perfusion. The level of human Fc in tissue homogenate extracts from different organs was measured by ELISA.

**Tumor growth and treatments**. A20 cells ($3 \times 10^6$) were subcutaneously injected into the right flank of Balb/c mice. Mice were treated intravenously with 20 μg IFNα-anti-PD-L1. To block PD-1 signaling, mice were treated intravenously with 100 μg anti-PD-1 (4H2) for twice a week, 1 day before IFNα-anti-PD-L1 treatment. For CD8[+] T-cell depletion, 200 μg anti-CD8 (TIB210) was injected i.p. 1 day before IFNα-anti-PD-L1 treatment. To block type I IFN signaling, 100 μg of anti-IFNAR1 (MAR1-5A3) was injected i.t. 1 day before IFNα-anti-PD-L1 treatment. MC38 cells ($5 \times 10^5$) were subcutaneously injected into the right flank of C57BL/6 mice. Mice

were treated intravenously with 25 µg IFNα-anti-PD-L1 twice. Tumor volumes were measured twice weekly and calculated as (Length × Width × Height/2). To block lymphocyte trafficking, mice were intraperitoneally injected with 25 µg FTY720, and 20 µg of FTY720 was administered every other day to maintain the blockade.

**Measurement of IFN-γ-secreting T cells by ELISPOT assay.** Draining lymph nodes (LNs) or spleen from tumor-bearing mice were isolated and single-cell suspensions were prepared. A20 tumor cells were irradiated with a single dose of 60Gys (10 Gys/min for 6 min). The spleen or LN cells were co-cultured with irradiated tumor cells at a ratio of 4:1 for 48 h. The IFN-γ production was determined with an IFN-γ ELISPOT assay kit according to the manufacturer's protocol (BD Biosciences). The cytokine spots were enumerated with the ImmunoSpot Analyzer (CTL).

**Ex vivo culture and function assays.** Tumor tissues were collected, cut into small pieces, and re-suspended in digestion buffer (RPMI-1640 medium with 1 mg/ml type IV collagenase and 100 µg/ml DNase I). Tumors were digested for 45 min at 37 °C, then passed through a 70 µm cell strainer to make single-cell suspensions. $CD8^+$ T cells and DCs ($MHCII^+$ $CD11c^+$) were sorted by FACS. T cells, DCs, and irradiated tumor cells were cultured at a ratio of 10: 1: 2.5 in the presence of IFNα-Fc (2 ng/ml) or anti-PD-L1 (10 µg/ml). Three days later, the supernatants were harvested and IFN-γ levels were measured by CBA.

**Patient data analysis.** An online database Kaplan–Meier plotter (http://kmplot.com/analysis/) was used to generate the survival plot[36]. Affymetrix ID for IFNA1 is 208344_x_at. For IFNA1 transcript abundancy, original data was obtained from a previous study[37]. Patients treated with anti-PD-1 (pembrolizumab or nivolumab) were grouped as progressive disease (PD, $n = 29$) or non-progressive disease (NPD, including complete response, partial response, and stable disease, $n = 36$). Relative IFNA1 transcript abundances were scored and compared.

**Statistical analysis.** Data are shown as the mean ± SEM. Statistical analyses were compared using an unpaired Student's two-tailed $t$-test. Analyses were performed using GraphPad Prism version 5.0 (GraphPad Software). Statistically significant differences of $p < 0.05$, $p < 0.01$ and $p < 0.001$ are noted with *, **, and ***, respectively.

## Data availability

The data supporting the findings of this study are available within the article and its Supplementary Information files and from the corresponding authors on reasonable request.

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

## Acknowledgements

We are grateful to Dr. Mingzhao Zhu for helpful suggestions and comments on the project. We thank Daryl Harmon for excellent editing. We thank Shuo Guo, Hui Su and Junfeng Hao for technical assistance, and Yuanyuan Chen and Zhenwei Yang for technician help with BLI experiments in IBP. This work was in part supported by the Cancer Prevention and Research Institute of Texas to Y.X.F., We appreciate the funding from Chinese Academy of Sciences (XDA12020212) and National Key R&D Program of China (2016YFC1303405) to H. Peng, and China Postdoctoral Science Foundation (No. 2016M600139) to Y. Liang.

## Author contributions

Y.L., J.G., H.T., H.P., and Y.X.F. conceptualized the project and designed experiments. Y.L., J.G., and H.T. performed experiments. X.Q., Z.Y., J.S., and Y.H. performed or contributed to specific experiments. Y.L., H.T., and J.G., analyzed data. Y.L., H.T., L.X., H.P.,

and Y.X.F. contribute to manuscript preparation. H.X., Z.R., Y.B., Z.S., and H.D. provided technical or material supports. H.P., H.T., and Y.X.F. supervised the project.

**Additional information**

**Competing interests:** The authors declare no competing interests.

