## [Peer Review File · Nature Communications]

Reviewers' Comments:

Reviewer #1:

Remarks to the Author:

This article describes successful preclinical studies using an anti-PD-L1-IFN-alpha fusion construct, resulting in strong antitumor activity. Remarkably, the construct works both when cancer cells express PD-L1 or are engineered to be deficient of this target. The antitumor activity results in an immune memory response, and does not negatively impact T cell responses to the cancer.

Major comments:

What evidence supports the first statement in the Abstract: "Most patients remain unresponsive to intensive PD therapy despite the presence of tumor infiltrating lymphocytes."? In fact, most data suggest that most patients not responding to anti-PD-1 therapy have absence of baseline T cell infiltrates, as opposed to having them and not responding.

The use of IFN-alpha2b for the treatment of melanoma and renal cell carcinoma was based on hypothesized direct antiproliferative effects but also immune modulation; therefore, it is hard to support the statement in the Introduction "Clinically, type I interferons (IFNs) have been approved by the FDA for the treatment of lymphoma and melanoma due to their pro-apoptosis and anti-proliferation effects on tumor cells, but their role in antitumor immunity has been unappreciated until recently^{9, 10}." I suggest deleting it.

The authors do not explain well that they seem to use an IFN-alpha fused to a Fc in their studies. This is detailed in the Figures and Methods, but omitted from the main text or figure legends.

Minor comments:

The article is plagued with laboratory jargon and incomplete descriptions, such as "PD therapy" instead of "PD-1 blockade therapy", for example, or "IFN" without stating which type of IFN.

Reviewer #2:

Remarks to the Author:

Overall this is an interesting paper reporting on newly engineered biomolecule encompassing a PD-L1 blocking antibody and murine IFNalpha4. A logical and properly controlled series of experiments shows some of the mechanisms at play in the efficacy against various transplantable mouse tumor models. Immunotherapy works intratumorally and intravenously given and mainly requires IFNAR expression on non-tumor cells. Interestingly further blockade of PD-1 enhances efficacy suggesting insufficient blockade. With regard to novelty type I IFN and immunostimulatory monoclonal antibodies has been already touched several times but this is quite a creative approach and the mechanism behind is well explored.

Comments:

1. What about effects of IFN α on T cells themselves. There is ample evidence that IFNAR $^{-/-}$ T cells totally underperform and direct effects on human CD8 T cells have been described. This should be discussed because T cells upon activation also express PD-L1. Effect of the compound on T cell activation should be tested and the literature on direct effects of IFN α on CTLs and CTL precursors should be referred to.
2. Could PD-L1 and IFNAR crosstalk on the plasma membrane or intracellularly. It is intriguing that the heterodimer works better. PD-L1 has been recently shown to signal in an antiapoptotic fashion on tumor cells (Group of David Escors).
3. Definitive evidence for homing of the construct to the tumor microenvironment should be

provided by bioluminescence of radioactivity (scintigraphy).

4. Page 8: There is no data about the IFN α specific activity of the IFN-anti-HBs molecules. As can be seen in Fig 2d, the specific activity of each fusion protein is different with significant differences in the EC50. To support the conclusion that the IFN-anti-PD-L1 is less toxic than IFN-anti-HBs, the authors should demonstrate that the specific activity is similar.

5. Is there any effect on tumor vasculature? This should be addressed.

Minor:

6. Intratumoral approaches with type I IFN for mouse are reported in the literature including viral vectors and direct intratumoral injection. Reference to this literature should be given. i.e: Clin Cancer Res. 2016 Aug 1;22(15):3791-800;

Int J Cancer. 2011 Jan 1;128(1):105-18; J Immunol. 2014 Oct 15;193(8):4254-60.

7. Clinical trials combining IFN β and PD-1 agents are completed or ongoing and should be referred: i.e Oncotarget. 2017 Apr 13;8(41):71181-71187

8. Page 7: In Figure 2e a control group is lacking (IFN-Fc). The antitumor data of this group is reported in Fig 1f but it could be interesting to see also the data in this panel.

9. Page 8: The sentence IFN-anti-PD-L1 fusion protein accumulated in tumor but not in normal tissue is misleading. The concentration in tissue is higher than in liver and kidney but the authors detect the protein in these organs.

10. A description of the biotechnology strategy to heterodimerize should be presented in methods and perhaps illustrated with a graph.

Reviewer #3:

Remarks to the Author:

Targeting type I interferon tumor entry by anti-PD-L1 creates feedforward antitumor responses to overcome innate and adaptive resistance is an interesting study in which Liang, Tang, and Guo et al., have developed an anti-PD-L1/type I IFN fusion construct (anti-PD-L1-IFN) which they have demonstrated effectively targets to tumours using pre-clinical mouse models and has significant anti-tumor effect even against larger tumors. While the study itself is interesting, there are several technical issues and a lack of mechanistic detail which has been discussed below. There are also issues relating to novelty as well. In my opinion, I believe this article would benefit from the suggestions made below.

Major comments:

1. The authors should discuss the study "Immune Cell-Poor Melanomas Benefit from PD-1 Blockade after Targeted Type I IFN Activation" by Bald et al Cancer Discovery 2014. They previously have demonstrated similar principles such as those demonstrated throughout this study, but in greater detail – however, they did not employ a fusion construct between anti-PD-L1 and type I IFN within their study. This prior work compromises the novelty.

2. In Figure 1C the authors demonstrate that there are functional differences between T cells within the lymph nodes of small versus large tumors. This is by itself a nice experiment, however, it is not alone convincing of the point they are trying to convey. In my opinion, the authors should compare the absolute number of T cells and other immune cells within the draining lymph node at the same time point, and also within the tumors themselves. It would also be very nice if the authors could sort and stimulate T cells from within the tumors themselves and determine whether they are functionally capable of producing different levels of cytokines.

3. In Figure 1E the authors have displayed survival data rather than tumor growth measurements. For obvious reasons, this is not an appropriate surrogate for tumor growth data and the authors should be encouraged to include the growth measurements.

4. Within Figure 1F, the authors compare the effect that i.t. and i.v. injection of IFN-Fc and argue that i.t. delivery is the only way to achieve a delay in tumor growth. This is a bit misleading, as the authors administer the same amount of IFN-Fc i.t. or i.v., meaning that the serum concentration would never actually reach the same concentration as observed within the tumor. The authors should be encouraged to carry out more detailed pharmacokinetic analyses to compare the effects

of increasing the serum IFN concentration and what effect that has on tumor growth.

5. The rationale behind the experiment contained within Figure 2d should be explained in greater detail.

6. Figure 2E and Figure 2F; In order for this comparison to make sense, the authors must ensure that they are able to achieve serum concentrations equivalent to that achieved via intra-tumor delivery. If this is not possible due to toxicity, the authors must carry out studies in which they test varying (increasing) concentrations and show the effect that this has on tumor growth rate.

7. Figure 3F; the IFN-Fc+a-PD-L1 and IFN-a-PD-L1 groups are not directly comparable for the reasons stated above.

8. Within Figure 3 the authors demonstrate the administration of their fusion construct increases PD-L1 expression on tumor cells, and IFNAR expression on leukocytes. These are interesting observations, but the biology underlying this effect has not been tested. Does the fusion construct induce type I IFN-mediated apoptosis, increasing T cell accumulation and in turn promoting IFN γ -mediated PD-L1 upregulation? The latter effect would also reasonably account for the increase in IFNAR+ CD45+ cells. The authors need to more completely interrogate the mechanism of action.

Minor comments:

1. "Systemic delivery of type I IFNs usually has limited antitumor activities and severe side effects" – This statement should be modified to reflect the fact that achieving serum concentrations of type I IFN high enough to have a potentially clinically-meaningful effect is usually associated with dose-limiting toxicities.

2. The authors should discuss the purity of the protein product preparations used for treatment.

3. "Since type I IFNs are the most potent cytokines including PD-L1" this needs to be referenced, there are certainly studies which have demonstrated IFN γ is more effective.

4. The manuscript would benefit greatly from proofreading. There are many spelling and grammatical issues throughout.

Reviewer #4:

Remarks to the Author:

The manuscript presents interesting and, in some instances, surprising results obtained with 2 different formats (a heterodimer and a homodimer) of anti PD-L1 scFv-IFN α fusion proteins with a human IgG1 Fc. The heterodimer format (achieved using knobs-into-holes technology) is a simple bispecific, with one half of the Fc fused to the scFv and the other fused to IFN α ; in the homodimer format, both arms are fusions of IFN α -scFv-Fc. Hence the fusion proteins are different in many ways. The heterodimer is monomeric for antibody and cytokine whilst the homodimer is dimeric, having the potential for increased functional affinity and cross linking in addition to being higher MW. There is also potential steric influence of the IFN α -scFv fusion on biological effects of either the antibody or the cytokine arm, and the stability of the resulting fusion proteins could differ enormously. The fusion proteins are the core of the work and their full characterisation should be reported in the manuscript. Information should include: yield (before and after purification, purity / stability / breakdown (SDS-PAGE and FPLC); affinity for PD-L1 (e.g. by SPR or ELISA; although the authors use the term I cannot see how the affinities were obtained or what they are); biological activity, PK and biodistribution. For examples of the types of data required (albeit with a different cytokine-scFv fusion) see Figs 1 and 2 in Hemmerle and Neri *Cancer Immunol Res*; 2(6); 559–67.

Reviewers' comments:

Reviewer #1 (Expertise: Check-point therapy, Remarks to the Author):

This article describes successful preclinical studies using an anti-PD-L1-IFN-alpha fusion construct, resulting in strong antitumor activity. Remarkably, the construct works both when cancer cells express PD-L1 or are engineered to be deficient of this target. The antitumor activity results in an immune memory response, and does not negatively impact T cell responses to the cancer.

Major comments:

1. What evidence supports the first statement in the Abstract: “Most patients remain unresponsive to intensive PD therapy despite the presence of tumor infiltrating lymphocytes.”? In fact, most data suggest that most patients not responding to anti-PD-1 therapy have absence of baseline T cell infiltrates, as opposed to having them and not responding.

We thank the reviewer for pointing out the inaccurate statement. We have revised it to “Many patients remain unresponsive to intensive anti-PD-1/PD-L1 therapy, despite the presence of tumor-infiltrating lymphocytes.”

2. The use of IFN-alpha2b for the treatment of melanoma and renal cell carcinoma was based on hypothesized direct antiproliferative effects but also immune modulation; therefore, it is hard to support the statement in the Introduction “Clinically, type I interferons (IFNs) have been approved by the FDA for the treatment of lymphoma and melanoma due to their pro-apoptosis and anti-proliferation effects on tumor cells, but their role in antitumor immunity has been unappreciated until recently^{9, 10.}” I suggest deleting it.

We have deleted the statement as suggested by the reviewer.

3. The authors do not explain well that they seem to use an IFN-alpha fused to a Fc in their studies. This is detailed in the Figures and Methods, but omitted from the main text or figure legends.

We have now included a detailed description of the IFN α -Fc fusion in the text (Yellow highlight on page 5, line 90).

Minor comments:

4. The article is plagued with laboratory jargon and incomplete descriptions, such as “PD therapy” instead of “PD-1 blockade therapy”, for example, or “IFN” without stating which

type of IFN.

We thank the reviewer's reminding for using complete terminology. We have corrected these descriptions, using "PD-L1/PD-1 blockade therapy" instead of "PD therapy", and using "IFN α " or "IFN γ " instead of "IFN".

Reviewer #2 (Expertise: Antigen presentation, DC activation, check-point therapy, Remarks to the Author):

Overall this is an interesting paper reporting on newly engineered biomolecule encompassing a PD-L1 blocking antibody and murine IFN α 4. A logical and properly controlled series of experiments shows some of the mechanisms at play in the efficacy against various transplantable mouse tumor models. Immunotherapy works intratumorally and intravenously given and mainly requires IFNAR expression on non-tumor cells. Interestingly further blockade of PD-1 enhances efficacy suggesting insufficient blockade. With regard to novelty type I IFN and immunostimulatory monoclonal antibodies has been already touched several times but this is quite creative approach and the mechanism behind is well explored.

Comments:

1. What about effects of IFN α on T cells themselves. There is ample evidence that IFNAR $^{-/-}$ T cells totally underperform and direct effects on human CD8 T cells have been described. This should be discussed because T cell upon activation also express PD-L1. Effect of the compound on T cell activation should be tested and the literature on direct effects of IFN α on CTLs and CTL precursors should be referred to.

We thank the reviewer for the supportive comments. We have cited the literatures that showed type I IFN could directly promote the proliferation and cytokine secretion of T cells (Page 11, line 222).

We have done in vitro assays to show that both IFN α -Fc and IFN α -anti-PD-L1 fusion proteins could active murine CD8⁺ T cells efficiently (**Figure R1a**). To determine whether activation of IFNAR signaling on CD8⁺ T cells is critical for the tumor control, we transferred sorted WT or IFNAR1^{-/-} CD8⁺ T cells into tumor bearing Rag-1 mice and treated with IFN α -anti-PD-L1 protein. IFN α -anti-PD-L1 could still inhibit the tumor growth when IFNAR in CD8⁺ T cells is deficient, suggesting the IFNAR signaling on T cells play minor roles in tumor control (**Figure R1b**).

We have included these data in **Supplemental Figure 8**.

Figure R1. IFNAR signaling on CD8 T cells play minor roles in tumor control by IFN α -anti-PDL1. (a) Splenic CD8⁺ T cells were sorted and allocated into anti-CD3 (1 μ g/ml) coated 96-well plate, and cultured in the presence of different concentrations of IFN α -anti-PD-L1 or control proteins for 24 hrs. CD25 expression were measured by FACS. (b) B6.Rag1^{-/-} mice were inoculated with 3×10^5 MC38 cells. 1×10^6 sorted CD8⁺ T cells from WT or IFNAR1^{-/-} mice were injected intravenously into tumor bearing Rag1^{-/-} mice on day 7. Mice were treated with 25 mg IFN α -anti-PD-L1 on day 11 and 14. Tumor growth was measured twice a week.

2. Could PD-L1 and IFNAR crosstalk on the plasma membrane or intracellularly. It is intriguing that the heterodimer works better. PD-L1 has been recently shown to signal in an antiapoptotic fashion on tumor cells (Group of David Escors).

Yes, the study by Dr. Escors' Group showed that PD-L1 cell-intrinsic signaling protects cancer cells from interferon (IFN) cytotoxicity and promote tumor progression (Gato-Canas, M. et al, 2017, Cell Reports).

To address the potential synergy between PD-L1 and IFNAR signaling, we treated A20 cells with different fusion proteins and measured tumor cell death (**Figure R2**). The apoptosis induced by IFN α -anti-PD-L1 fusion protein is limited, and only slightly increased compared to IFN α -Fc alone treatment.

Figure R2. IFN α -anti-PD-L1 induces limited apoptosis in A20 cells. 2×10^5 A20 tumor cells were cultured in vitro with 10 pm of indicated proteins in the medium. 48 hours later, staining with annexinV-AF647 and PI was performed to distinguish necrotic (annexin-PI+), early apoptotic (annexin+PI-), and late apoptotic (annexin+PI+) cell populations. The percentage of total apoptotic cells was quantified for each sample as the sum of

early apoptotic and late apoptotic cells. Mitomycin (5 μ g/ml) treatment were used as positive control of apoptosis.

3. *Definitive evidence for homing of the construct to the tumor microenvironment should be provided by bioluminescence of radioactivity (scintigraphy).*

As suggested by the reviewer, we traced the protein homing by bioluminescence, which showed that fusion protein accumulated in tumor tissue efficiently (**Figure R3**). This data has been included in **Figure 3e**.

Figure R3. IFN α -anti-PD-L1 targets to the tumor tissues in vivo. IFN α -anti-PDL1 protein was conjugated with Cy5.5-maleimide. 25 μ g of CY5.5- IFN α -anti-PDL1 protein was injected into A20 tumor bearing mice. The protein accumulating within tumor was tracked by IVIS spectrum in vivo imaging system.

4. *Page 8: There is no data about the IFN α specific activity of the IFN-anti-HBs molecules. As can be seen in Fig 2d, the specific activity of each fusion protein is different with significant differences in the EC50. To support the conclusion that the IFN α -anti-PD-L1 is less toxic than IFN-anti-HBs, the authors should demonstrate that the specific activity is similar.*

We have compared the bioactivities of IFN α - α -HBs and IFN α - α -PD-L1 (**Figure R4**). There is no significant differences. We have included these data in **supplementary Figure 6a**.

Figure R4. IFN- α -HBs and IFN α -anti-PD-L1 have similar bioactivities. The bioactivity of IFN α - α -PD-L1 and control IFN α - α -HBs protein was measured by an antiviral infection biological assay.

5. *Is there any effect on tumor vasculature? This should be addressed.*

Previous studies have showed that local treatment with a high dose of type I IFN exerts antitumor effects through direct anti-angiogenic effects¹. We have stained tumor vasculatures after IFN α -anti-PD-L1 treatment (**Figure R5**). Interestingly, IFN α -anti-PD-L1 treatment significantly reduced tumor vasculatures in tumor tissues, especially in the center of the tumor. How the effects of IFN α -anti-PD-L1 on angiogenesis

contribute to CTL or CTL contributes to the change of vasculatures remains to be determined. However, this is out of the scope of the current study.

Figure R5. IFN α -anti-PD-L1 treatment inhibits angiogenesis. MC38 tumor-bearing mice were treated with 25 μ g α -PD-L1 or IFN α -anti-PD-L1 on day 14 and 17. Tumor tissues were collected and stained for CD31 on day 20.

Minor:

6. *Intratumoral approaches with type I IFN for mouse are reported in the literature including viral vectors and direct intratumoral injection. Reference to this literature should be given.i.e: Clin Cancer Res. 2016 Aug 1;22(15):3791-800;Int J Cancer. 2011 Jan 1;128(1):105-18; J Immunol. 2014 Oct 15;193(8):4254-60.*

We thank the reviewer for the comments. We have cited these papers (Page 6, line106).

7. *Clinical trials combining IFN β and PD-1 agents are completed or ongoing and should be referred: i.e Oncotarget. 2017 Apr 13;8(41):71181-71187*

We have cited these studies in the revision (Page 18, line374).

8. *Page 7: In Figure 2e a control group is lacking (IFN-Fc). The antitumor data of this groups is reported in Fig 1f but it could be interesting to see also the data in this panel.*

We have added the data of IFN-Fc (see Figure R6) to **Figure 2e**.

Figure R6. Treatment of IFN α -anti-PD-L1 fusion protein via i.t. injection inhibit the tumor growth. Balb/c mice were inoculated with 3×10^6 A20 cells. After tumor established, mice were treated with 20 μ g of control, anti-PD-L1, IFN α -Fc, or protein by i.t. (days 12 and 16) injection. Tumor size was measured per week.

9. *Page8: The sentence IFN-anti-PD-L1 fusion protein accumulated in tumor but not in normal tissue is misleading. The concentration in tissue is higher than in liver and kidney but the authors detect the protein in these organs.*

We have changed our statement for this results as suggested by the reviewer (Page 8, line 172).

10. *A description of the biotechnology strategy to heterodimerize should be presented in methods and perhaps illustrated with a graph.*

We have added a description of biotechnology strategy to heterodimer in methods (Page 20,

line 407), and revised the graph illustration of fusion protein construction in Figure 2a.

Reviewer #3 (Expertise: IFN, cancer immunotherapy, Remarks to the Author):

Targeting type I interferon tumor entry by anti-PD-L1 creates feedforward antitumor responses to overcome innate and adaptive resistance is an interesting study in which Liang, Tang, and Guo et al., have developed an anti-PD-L1/type I IFN fusion construct (anti-PD-L1-IFN) which they have demonstrated effectively targets to tumours using pre-clinical mouse models and has significant anti-tumor effect even against larger tumors. While the study itself is interesting, there are several technical issues and a lack of mechanistic detail which has been discussed below. There are also issues relating to novelty as well. In my opinion, I believe this article would benefit from the suggestions made below.

Major comments:

1. The authors should discuss the study “Immune Cell-Poor Melanomas Benefit from PD-1 Blockade after Targeted Type I IFN Activation” by Bald et al Cancer Discovery 2014. They previously have demonstrated similar principles such as those demonstrated throughout this study, but in greater detail – however, they did not employ a fusion construct between anti-PD-L1 and type I IFN within their study. This prior work compromises the novelty.

We have added a discussion in the revised manuscript (Page 16, line334).

The study by Bald et al. show that targeted activation of type I IFN system by peri-tumoral injection of poly (I:C) induces antitumor immune responses, and the effects can be further enhanced by a combination therapy with PD-1 blockade. Comparing to their study, our design has several advantages. First, peri-tumoral injection is not feasible to most patients in clinic. Our design that utilizing anti-PD-L1 as a carrier to deliver IFN α into tumor tissues overcome the targeting problem. Second, our fusion protein has significantly reduced the side effects due to the tumor targeting effects. Third, IFN α -anti-PD-L1 as protein drug is easier to be accepted than poly (I:C) clinically.

2. In Figure 1C the authors demonstrate that there are functional differences between T cells within the lymph nodes of small versus large tumors. This is by itself a nice experiment, however, it is not alone convincing of the point they are trying to convey. In my opinion, the authors should compare the absolute number of T cells and other immune cells within the draining lymph node at the same time point, and also within the tumors themselves. It would also be very nice if the authors could sort and stimulate T cells from within the tumors themselves and determine whether they are functionally capable of producing different levels of cytokines.

To find out whether there are functional differences between T cells within dLN of small

versus large tumor responding to PD-L1 blockade, we have done IFN γ and TNF α in vivo intracellular FACS staining on three days after PD-L1 blockade treatment. The results showed that both the frequency and absolute number of IFN γ^+ TNF α^+ effector T cells increased in small tumor after PD-L1 blockade, but there are no significant changes in large tumor (**Figure R7a and R7b**). Consistent with Figure 1c, these results showed that PD-L1 blockade induces robust T-cell activation in small tumors, while has limited effects on T cells in advanced tumors. PD-L1 blockade treatment also led to a trend of increased NK and macrophage in the dLN of small but not large tumor, while there is no significant change of other immune cells (**Figure R7e and R7f**).

In addition, we compared the functional status of T cells within tumor tissues. The frequency and absolute number IFN γ^+ TNF α^+ CD4 $^+$ T cells increased in small but not large tumor at 3 days after PD-L1 blockade treatment (**Figure R7c and R7d**). There is no significant change of other immune cells (**Figure R7e and R7f**). There is an increase in MDSC frequency within large tumor compared to small tumor, while the effector CD8 $^+$ T cells were reduced (**Figure R7g and R7h**), which may affect the responsiveness to PD-L1 blockade. Part of these data was included in **Supplement Figure 1**.

Figure R7. PD-L1 blockade activate the effector function of T cells in small tumor but not advanced tumor. Balb/c mice (n=4) bearing early stage tumors (<50 mm³) or advanced A20 tumors (>100 mm³) were treated intraperitoneally (i.p.) with 200 µg of anti-PD-L1. After 3 days, mice were i.p. injected with 250 µg Brefeldin A (BFA). 6 hours later, draining Lymph node and tumor tissues were collected and digested. FACS staining was performed. (a) The frequency of IFN γ ⁺TNF α ⁺CD4⁺ or CD8⁺ T cells within dLN, (b) The absolute number of IFN γ ⁺TNF α ⁺CD4⁺ or CD8⁺ T cells within dLN, (c) The frequency of IFN γ ⁺TNF α ⁺CD4⁺ or CD8⁺ T cells within tumor, (d) The absolute number of IFN γ ⁺TNF α ⁺CD4⁺ or CD8⁺ T cells within tumor, (e) The frequency of indicated cell subsets within dLN, (f) The absolute number of indicated cell subsets within dLN, (g) The frequency of indicated cell subsets within tumor, (h) The absolute number of indicated cell subsets within tumor tissues were shown. NK (F4/80⁻CD3⁻NK1.1⁺), macrophage (F4/80⁺), MDSC (CD11b⁺Gr-1⁺), Treg (CD4⁺Foxp3⁺), B cell (B220⁺).

3. In Figure 1E the authors have displayed survival data rather than tumor growth measurements. For obvious reasons, this is not an appropriate surrogate for tumor growth data and the authors should be encouraged to include the growth measurements.

We have added the tumor growth curve in **supplemental Figure 2b**.

4. Within Figure 1F, the authors compare the effect that i.t. and i.v. injection of IFN-Fc and argue that i.t. delivery is the only way to achieve a delay in tumor growth. This is a bit misleading, as the authors administer the same amount of IFN-Fc i.t. or i.v., meaning that the serum concentration would never actually reach the same concentration as observed within the tumor. The authors should be encouraged to carry out more detailed pharmacokinetic analyses to compare the effects of increasing the serum IFN concentration and what effect that has on tumor growth.

We thank reviewer for the comments. In this specific experiment, what we intent to show is that tumor control mainly depends on the interactions of IFN α with immune cells within tumor tissue but not peripheral. Consistent with our FTY720 blocking experiment (**Figure 6g**), intratumoral delivery of IFN α is far more potent than systemic delivery. We have revised our manuscript to better clarify these points (**Page 6, line 92**).

As suggested by the reviewer, we carried out a more detailed analysis to the pharmacokinetic (**Figure R8**). Although systemic delivery of a very high dose of IFN α -Fc (80 μ g) resulted in high levels of protein in serum, the level in tumor tissues was limited (**Figure R8a & R8b**) and there were no significant tumor control effect (**Figure R8c**). Interestingly, local delivery of a lower dose of IFN α -Fc (20 μ g) controlled tumor growth much better despite the level in serum was much lower (**Figure R8c & R8d**). These data suggested that type I IFN needs to target to tumor local for an optimal tumor control.

Figure R8. Pharmacokinetic study of IFN α -Fc. Different doses of IFN α -Fc were injected intratumorally (i.t.) or intravenously (i.v.) into A20 tumor bearing mice. Serum were collected at 15 min, 1 hrs, 6 hrs, 1 days, 3 days and 5 days post injection. Serum concentration of IFN α -Fc was measured by ELISA. Tumor growth was measured twice per week.

5. The rationale behind the experiment contained within Figure 2d should be explained in greater detail.

We have now explained more about the rationale of Figure 2d (Page 7, line 130).

6. Figure 2E and Figure 2F; In order for this comparison to make sense, the authors must ensure that they are able to achieve serum concentrations equivalent to that achieved via intra-tumor delivery. If this is not possible due to toxicity, the authors must carry out studies in which they test varying (increasing) concentrations and show the effect that this has on tumor growth rate.

One concern is that higher serum concentration of IFN α -Fc lead to better tumor control effect. We have compared the serum and tumor concentrations of IFN α -Fc after systemic or local delivery (please see our responses to Question #4). As showed in **Figure R8**, systemic delivery of IFN α -Fc resulted in high level of protein in serum but not in tumor tissue. In contrast, local delivery resulted in higher level of protein in tumor but lower level in serum. These data showed that tumor control mainly depends on IFN signaling in tumor tissues but not in peripheral.

As suggested by the reviewer, we have tested different concentrations of fusion protein treatment and monitored their antitumor effects (**Figure R9**). Systemic delivery of heterodimer inhibited tumor growth more efficiently at higher dose, but still weaker than local injection (**Figure R9a**). In contrast, the therapeutic effect of homodimer was weaker than heterodimer, even in much higher dose (**Figure R9b**). We have added these data into **supplementary Figure 4**).

Figure R9. Titration of fusion proteins for antitumor effects.

Balb/c mice (n=4 ~6) were inoculated with 3×10^6 A20 cells. After tumor

established, mice were treated with indicated doses of IFN- α -PDL1 heterodimer or homodimer protein by i.v. injection twice (day 11 and 15). Tumor size was measured twice per week.

7. Figure 3F; the IFN-Fc+a-PD-L1 and IFN-a-PD-L1 groups are not directly comparable for the reasons stated above.

For IFN-Fc+a-PD-L1 vs IFN α -anti-PD-L1 groups, we agree that it is hard to compare these two. The differences could be caused by differences in protein serum half-lives, overall biological activities, modes of actions, or targeting effects, et al. However, the purpose of this experiment was to have a practical comparison rather than direct comparison between fusion

protein and a simple mixture of equal mole of proteins. We meant to propose that IFN α -anti-PD-L1 but not IFN α -Fc could accumulate within tumor tissues, which express high level of PD-L1 for targeting. We have better clarified it in the revised manuscript. We thank the reviewer for pointing them out.

8. *Within Figure 3 the authors demonstrate the administration of their fusion construct increases PD-L1 expression on tumor cells, and IFNAR expression on leukocytes. These are interesting observations, but the biology underlying this effect has not been tested. Does the fusion construct induce type I IFN-mediated apoptosis, increasing T cell accumulation and in turn promoting IFN γ -mediated PD-L1 upregulation? The latter effect would also reasonably account for the increase in IFNAR⁺ CD45⁺ cells. The authors need to more completely interrogate the mechanism of action.*

This is a good point for us to further explore the functional mechanism of fusion proteins. To study if IFN α increase T cell accumulation and in turn promoting IFN γ -mediated PD-L1 upregulation, we inhibited T cell trafficking by FTY720, then measured PD-L1 upregulation by FACS. We found that PD-L1 was upregulated in the absence of T cells migrating into tumor tissues (**Figure R10a**). Next, we inoculated MC38 tumor into Rag-1 mice which have no T cells, and tested whether IFN- α -PD-L1 could still upregulate PD-L1. PD-L1 was significantly upregulated within tumor tissue after IFN- α -PD-L1 treatment (**Figure R10c**). These results suggested that IFN- α -PD-L1 upregulates PD-L1 directly, rather than through indirect effects. These data were added into **supplemental Figure 7**.

For IFNAR expression, we cannot rule out whether the differences were related to different cell populations or different levels of IFNAR on the same cell population. We have removed this data.

Figure R10. PD-L1 upregulation does not depend on T cell trafficking. (a) 3×10^5 MC38 were inoculated into B6 mice.

FTY720 was administered every other day from day 7. 25 μ g of IFN- α -PDL1 was injected into mice on day 8. Two days later, tumor tissues were harvested, and PD-L1 expression were determined by flow cytometry. (b) 3×10^5 MC38 were inoculated into Rag mice. 25 μ g of IFN- α -PDL1 was injected into mice on day 9. Two days later, tumor tissues were harvested, and PD-L1 levels were determined by flow cytometry.

Minor comments:

9. *“Systemic delivery of type I IFNs usually has limited antitumor activities and severe side effects” – This statement should be modified to reflect the fact that achieving serum concentrations of type I IFN high enough to have a potentially clinically-meaningful effect is*

usually associated with dose-limiting toxicities.

We thank the reviewer for pointing it out. We have revised the text as suggested by the reviewer (Page 6, line 101).

10. The authors should discuss the purity of the protein product preparations used for treatment.

We have run non-reducing and reducing SDS-PAGE (**Figure R11a**), and capillary electrophoresis (CE) (**Figure R11b**) to show the purity of the used proteins. All of the proteins have the purity larger than 95%. We have included these descriptions in methods and **Supplemental Figure 3a and 3b**.

Figure R11. Biochemical characterization of the fusion protein. (a) Fusion proteins were purified from culture supernatant in 293F cells, and were analyzed by non-reducing (left) and reducing (right) SDS-PAGE after purification. M, Marker; 1, α -PD-L1; 2, IFN α -Fc; 3, IFN α -anti-PD-L1 heterodimer; 4, homodimer; 5, IFN- α -HBs. (b) CE electropherograms of indicated proteins.

11. “Since type I IFNs are the most potent cytokines including PD-L1” this needs to be referenced, there are certainly studies which have demonstrated IFN gamma is more effective.

Both type I and II IFNs could upregulate PD-L1 expression potently. We have cited the papers (Page 9, line 179), and corrected into “Since type I IFNs are one of most potent cytokines inducing PD-L1”

12. The manuscript would benefit greatly from proofreading. There are many spelling and grammatical issues throughout.

We have done proofreading by professional language editing service.

Reviewer #4 (Fusion Ab development, Remarks to the Author):

The manuscript presents interesting and, in some instances, surprising results obtained with 2 different formats (a heterodimer and a homodimer) of anti PD-L1 scFv-IFN α fusion proteins

with a human IgG1 Fc. The heterodimer format (achieved using knobs-into-holes technology) is a simple bispecific, with one half of the Fc fused to the scFv and the other fused to IFN α ; in the homodimer format, both arms are fusions of IFN α -scFv-Fc. Hence the fusion proteins are different in many ways. The heterodimer is monomeric for antibody and cytokine whilst the homodimer is dimeric, having the potential for increased functional affinity and cross linking in addition to being higher MW. There is also potential steric influence of the IFN α -scFv fusion on biological effects of either the antibody or the cytokine arm, and the stability of the resulting fusion proteins could differ enormously. The fusion proteins are the core of the work and their full characterisation should be reported in the manuscript. Information should include:

1. yield (before and after purification, purity / stability / breakdown (SDS-PAGE and FPLC);

We have added the results of yield in the method section (**Page 20, line 398**). Furthermore, we have run SDS-PAGE and capillary electrophoresis (CE) to show the purity of protein, which is >95% pure (please refer to **Figure R11a and R11b**).

To evaluate the stability, proteins were incubated with mouse serum at 37°C for various time. The activity of proteins was detected by anti-viral infection bioassay. Heterodimer was found more stability than homodimer (**Figure R12**). We have included this data into **Supplemental Figure 5**.

Figure R12. The stability of IFN α -anti-PD-L1 fusion protein. To compare the stability, IFN- α -PDL1 homodimer (a) or heterodimer (b) were added into mouse serum and incubate at 37°C for indicated time. The activity of proteins was detected by IFN anti-viral infection bioassay.

2. affinity for PD-L1 (e.g. by SPR or ELISA; although the authors use the term I cannot see how the affinities were obtained or what they are);

We have measured the binding affinity to PD-L1 by using Biolayer Interferometry (BLI) (**Figure R13**). The data is shown in **supplemental Figure 3c**.

Figure R13. The binding affinity of proteins to PD-L1. Binding curves of α -PD-L1, IFN α -anti-PD-L1 heterodimer or IFN α -anti-PD-L1 homodimer to immobilized PD-L1-Fc-biotin. The constants were determined by dynamic-analysis model. $K_D = k_d/k_a$ (d, dissociation; a, association).

3. biological activity,

We have added IFN biological activity data including control IFN α -anti-HBs protein. See **Figure. 2d** and **Supplemental Figure. 6a**.

4. PK and biodistribution. For examples of the types of data required (albeit with a different cytokine-scFv fusion) see Figs 1 and 2 in Hemmerle and Neri *Cancer Immunol Res*; 2(6); 559–67.

The PK results have been added in **Fig. 2h** and **2i**, which show that heterodimer accumulates in tumor tissues with a much higher level comparing to homodimer (**Fig. 2h**). In addition, the serum half-life of heterodimer was significantly longer (**Fig. 2i**).

The biodistribution of IFN α - α -PDL1 was analyzed in **Fig.3d**. High concentration of IFN α -anti-PD-L1 fusion protein was retained within tumor even on day 5 post i.v. injection, and much less protein was found in normal tissues. We have also added the bioluminescence data showing that fusion protein accumulates in tumor definitely (**Fig.3e**).

References

1. Spaapen, R.M. et al. Therapeutic activity of high-dose intratumoral IFN-beta requires direct effect on the tumor vasculature. *Journal of immunology* **193**, 4254-4260 (2014).

Reviewers' Comments:

Reviewer #1:

Remarks to the Author:

The authors have addressed the reviewer's comments and improved the article.

Reviewer #2:

Remarks to the Author:

My previous comments are successfully addressed.

Reviewer #3:

Remarks to the Author:

The authors of the study "Targeting type I interferon tumor entry by anti-PD-L1 creates antitumor responses to overcome innate and adaptive resistance" have gone to great lengths to address reviewer's comments, however, additional issues are present within the study that might need to be addressed prior to acceptance.

1. The link between the first and second sentences of the results section is missing. The authors should show that smaller A20 tumors contain a higher CD8+ T cell to tumor cell ratio. While this would likely be simple and Supplemental data at most, it would complement their subsequent analyses.
2. Line 88 to 89 "these data raised the possibility that insufficient T-cell activation may be responsive for tumor resistance to checkpoint blockade". Is it not more likely that the T cells are simply more exhausted? The cytokine data certainly suggests this. However, I am not aware of any study that has shown that type I IFN can relieve T cell dysfunction/exhaustion. This might be an interesting discussion point.
3. Line 95 to 96 "Advanced tumors were treated with combination therapy of PD-L1 blockade and IFN α -Fc". They were not advanced – they were the same size as the "small tumors" treated in Figure 1A – there needs to be some consistency here, otherwise this is misleading.
4. Also, the authors need to discuss how their fusion construct (IFN-PD-L1 compares to combination anti-PD-L1 and IFN-Fc, better/worse/as good?).
5. For Figure 1F, the tumor growth data should be introduced into the main figure – it is more informative than the survival data.
6. Relating to Figure 1F, the authors should consider including the following or something along these lines: "It is possible that IV administration of the fusion product is less effective, because it is not possible to reach sufficient serum concentrations to enable intra-tumor accumulation to enable comparable accumulation achieved via IT injection".
7. Relating to Figure 2e and other figures. Regarding the anti-PD-L1/IFN-Fc controls included within each experiment, are they administered at equi-molar concentrations as the fusion product? This is probably important, but difficult to assess. The authors should consider discussing this limitation or address it experimentally.
8. 1. In Figure 2f, why does anti-PD-L1 lose its efficacy when administered IV? (ie, why is this so different from the result in Figure 1a?)

9. Related to the section beginning on line 165 "IFN α -armed anti-PD-L1 shows maximal antitumor activities with minimal toxicities", the authors have gone to great lengths to demonstrate that the fusion construct is safe in mice. However, it would be a good idea if the authors should discuss that immune-related adverse events are extremely difficult to assess within mice. This does not detract from their analysis, however, I feel it is fair to mention.

10. Line 202 "(data not show)". The data should be included – supplemental if necessary.

11. Figure 4f is missing a negative control, PD-L1ko tumor cells in PD-L1ko mice.

Reviewer #4:

Remarks to the Author:

The authors have made substantial efforts to address reviewers comments and the manuscript is much improved.

REVIEWERS' COMMENTS:

Reviewer #1 (Remarks to the Author):

The authors have addressed the reviewer's comments and improved the article.

We are glad to hear that we have addressed the reviewer's comments, and the manuscript has been improved.

Reviewer #2 (Remarks to the Author):

My previous comments are successfully addressed.

We are glad to hear that we have addressed the reviewer's comments.

Reviewer#3:

The authors of the study ““Targeting type I interferon tumor entry by anti-PD-L1 creates antitumor responses to overcome innate and adaptive resistance” have gone to great lengths to address reviewer's comments, however, additional issues are present within the study that might need to be addressed prior to acceptance.

1. The link between the first and second sentences of the results section is missing. The authors should show that smaller A20 tumors contain a higher CD8+ T cell to tumor cell ratio. While this would likely be simple and Supplemental data at most, it would complement their subsequent analyses.

We appreciate the review's suggestion. We did compare the CD8+ T cells ratio within tumor, and found that smaller A20 tumors contain higher ratio of CD8+ T cells. We have added the data into the manuscript. Please see line 76 and Supplemental Fig.1a.

2. Line 88 to 89 “these data raised the possibility that insufficient T-cell activation may be responsive for tumor resistance to checkpoint blockade”. Is it not more likely that the T cells are simply more exhausted? The cytokine data certainly suggests this. However, I am not aware of any study that has shown that type I IFN can relieve T cell dysfunction/exhaustion. This might be an interesting discussion point.

This is a good point. We have revised the statement as “These data suggest that the T cells might be more exhausted in advanced tumor, and could not be activated efficiently by checkpoint blockade alone.” (line 85)

3. Line 95 to 96 “Advanced tumors were treated with combination therapy of PD-L1 blockade and IFN α -Fc”. They were not advanced – they were the same size as the “small tumors” treated in Figure 1A – there needs to be some consistency here, otherwise this is misleading.

We apologized for the confusion and have revised it in the main text (line92). The size of tumors (60~75mm³) used here were larger than “small tumors”, but indeed smaller than “advanced tumor” in Fig1B. However, the tumors were still resistant to PD-L1 blockade alone, while the combination therapy of PD-L1 blockade with IFN α -Fc resulted in complete tumor eradication.

We have corrected our wording in the main text (line92).

4. Also, the authors need to discuss how their fusion construct (IFN-PD-L1 compares to combination anti-PD-L1 and IFN-Fc, better/worse/as good?).

We discussed in the discussion part (line 335).

5. For Figure 1F, the tumor growth data should be introduced into the main figure – it is more informative than the survival data.

We have introduced the tumor growth data into the main figure 1e, and put the survival data into the supplemental figure 2b, and we revised the main text (line 98).

6. Relating to Figure 1F, the authors should consider including the following or something along these lines: “It is possible that IV administration of the fusion product is less effective, because it is not possible to reach sufficient serum concentrations to enable intra-tumor accumulation to enable comparable accumulation achieved via IT injection”.

We appreciate the reviewer’s comments and have included these into the main text (line 101).

7. Relating to Figure 2e and other figures. Regarding the anti-PD-L1/IFN-Fc controls included within each experiment, are they administered at equi-molar concentrations as the fusion product? This is probably important, but difficult to assess. The authors should consider discussing this limitation or address it experimentally.

The molecular size of anti-PD-L1-Fc, IFN-Fc and IFN α -anti-PD-L1 heterodimer are

about 100kd, while IFN α -anti-PD-L1 homodimer is about 140kd. In Fig.2e and 2f, we injected same microgram weight of proteins, and the molar concentration of single of anti-PD-L1 or IFN-Fc is doubled compared to that of IFN α -anti-PD-L1. Moreover, the molar concentrations of anti-PD-L1/IFN-Fc of homodimer is about 1.4 fold compared to the same microgram weight of IFN α -anti-PD-L1 heterodimer. Even in such situation, the heterodimer showed similar tumor control effect via i.t. and via i.v., compared to homodimer.

We did compare the therapeutic effect of control and fusion proteins in an equi-molar concentrations of anti-PD-L1/ IFN-Fc in Fig.3f, and heterodimer still achieved the therapeutic effect.

8. 1. In Figure 2f, why does anti-PD-L1 lose its efficacy when administered IV? (ie, why is this so different from the result in Figure 1a?)

In Figure 2F, 20 μ g of anti-PD-L1 was injected to compare with the same microgram IFN α -anti-PD-L1 fusion protein, and it could not control tumor efficiently. While in Figure 1a, 200 μ g of anti-PD-L1 was administrated to control small A20 tumor effectively.

9. Related to the section beginning on line 165 “IFN α -armed anti-PD-L1 shows maximal antitumor activities with minimal toxicities”, the authors have gone to great lengths to demonstrate that the fusion construct is safe in mice. However, it would be a good idea if the authors should discuss that immune-related adverse events are extremely difficult to assess within mice. This does not detract from their analysis, however, I feel it is fair to mention.

It is hard to evaluate the immune-related adverse effect with murine model, because usually mice are less sensitive to the drug than human. Here we used a higher dose (μ g) of IFN α -anti-PD-L1 fusion protein to evaluate the toxicity within mice. We added this point in line 163.

10. Line 202 “(data not show)”. The data should be included – supplemental if necessary.

We have included the related data into Supplemental Fig.8.

11. Figure 4f is missing a negative control, PD-L1ko tumor cells in PD-L1ko mice.

Previous study has shown that PD-L1 KO tumor could not be established in PD-L1 KO mice¹. It may be not possible to include this as negative control.

Reviewer #4 (Remarks to the Author):

The authors have made substantial efforts to address reviewers comments and the manuscript is much improved.

We are glad to hear that we have addressed the reviewer's comments.

1 Lau, J. *et al.* Tumour and host cell PD-L1 is required to mediate suppression of anti-tumour immunity in mice. *Nature communications* **8**, doi:Artn 1457210.1038/Ncomms14572 (2017).